# The carbon dioxide removal potential of cement and lime kiln dust via ocean alkalinity enhancement

Gunter Flipkens[1*], Greet Lembregts[1] and Filip J.R. Meysman[1]

[1]Geobiology, Department of Biology, University of Antwerp, Antwerp, Belgium

*Correspondence to*: Gunter Flipkens (Gunter.Flipkens@uantwerpen.be)

**Keywords**: ocean liming, ocean alkalinity enhancement, CDR, cement kiln dust, lime kiln dust

**Abstract.** Ocean alkalinity enhancement (OAE) is a proposed method for atmospheric carbon dioxide removal (CDR), and involves the addition of alkaline minerals to surface waters to elevate seawater alkalinity and enhance atmospheric $CO_2$ storage. Cement kiln dust (CKD) and lime kiln dust (LKD) are alkaline side streams from the cement and lime industry that

have OAE potential due to their widespread availability and fine particle size. Here, we evaluated the dissolution kinetics, $CO_2$ sequestration potential, and ecological risks of CKD and LKD by means of laboratory dissolution experiments. A reactive fraction (~25 % in LKD and ~29 % in CKD) dissolved rapidly within 24 hours, with most dissolution occurring within the first hour. Dissolution provided a concomitant alkalinity release that was higher for LKD (up to $8.0 \pm 0.5$ mmol alkalinity per g) than CKD ($2.4 \pm 0.2$ mmol g$^{-1}$), thus providing a sizeable $CO_2$ sequestration capacity for LKD ($297 \pm 20$ g $CO_2$ per kg) and

CKD ($88 \pm 6$ g $CO_2$ per kg). Based on current industrial production rates, this translates into global CDR potentials of up to $8.7 \pm 0.6$ Mt $CO_2$ yr$^{-1}$ for LKD and $25 \pm 2$ Mt $CO_2$ yr$^{-1}$ for CKD. These estimates suggest that both materials could be viable OAE feedstocks, although further testing under conditions that more closely mimic natural coastal conditions is needed. Furthermore, we hypothesize that the substantial residual calcite content of LKD (~54 %) and CKD (~37 %) may provide additional sequestration via metabolic dissolution in marine sediments. However, kiln dust deployment will generate elevated

turbidity levels that may exceed environmental thresholds, underscoring the need for carefully designed application strategies to minimize local ecological impacts.

## 1 Introduction

As global temperatures continue to rise, the urgency for climate mitigation increases (IPCC, 2023). In addition to substantial cuts in greenhouse gas emissions, gigaton-scale atmospheric carbon dioxide removal (CDR) is needed to meet the goals of the

Paris Agreement (Rockström et al., 2017; Minx et al., 2018; IPCC, 2023). One proposed CDR method is mineral-based ocean alkalinity enhancement (OAE), which offers considerable removal potential and long-term $CO_2$ storage (Kheshgi, 1995; Renforth and Henderson, 2017). Mineral-based OAE targets the addition of specific basic minerals to the surface ocean, which

release alkalinity upon dissolution, thereby stimulating oceanic $CO_2$ uptake (Renforth and Henderson, 2017; Geerts et al., 2025).

A variety of minerals have been proposed as feedstock for mineral-based OAE, and approaches largely fall into two categories. So-called "enhanced weathering" techniques target the addition of natural rock material containing silicates (e.g. olivine $Mg_2SiO_4$) or carbonates (e.g. calcite $CaCO_3$) to sediments, which then slowly dissolve up or in the seabed over a time scale of years to decades (Montserrat et al., 2017; Flipkens et al., 2023; Dale et al., 2024; Fuhr et al., 2025; Geerts et al., 2025). In contrast, "ocean liming" procedures involve the addition of industrially processed minerals such as slaked lime ($Ca(OH)_2$) or

brucite ($Mg(OH)_2$), thus targeting a much faster alkalinity release upon dissolution in the water column over a time scale of minutes to hours (Renforth et al., 2013; Renforth and Henderson, 2017; Caserini et al., 2021; Foteinis et al., 2022; Kitidis et al., 2024). Slaked lime is produced by calcining limestone ($CaCO_3$) to form quicklime ($CaO$) (Eq. (1)), which is then hydrated (Eq. (2)) (Moras et al., 2022).

$$CaCO_3 \rightarrow CaO + CO_2 \qquad (1)$$

$$CaO + H_2O \rightarrow Ca(OH)_2 \qquad (2)$$

Upon dissolution in seawater, slaked lime releases alkalinity and binds $CO_2$ in the form of bicarbonate ($HCO_3^-$).

$$Ca(OH)_2 + 2CO_2 \rightarrow Ca^{2+} + 2HCO_3^- \qquad (3)$$

Ocean liming has the benefit of rapid seawater alkalinization upon deployment, and offers the potential to remove gigatons of atmospheric $CO_2$ annually, with ample global reserves to support deployment (Caserini et al., 2022; Foteinis et al., 2022).

However, large-scale deployment would require a substantial increase in limestone mining and lime production, carrying significant economic and environmental costs (Caserini et al., 2021; Foteinis et al., 2022). As a result, industrial by-products and mine tailings are attracting a growing interest for CDR applications, thanks to their lower processing costs, and alignment with circular economy principles (Bullock et al., 2021; Bullock et al., 2022; Moras et al., 2024).

The cement and lime industries are among the largest mineral production sectors globally, producing approximately 4.1 Gt of

cement and 0.42 Gt of lime annually (CEMBUREAU, 2024; USGS, 2025). Both rely on the high-temperature calcination of limestone in kilns to produce quicklime and cement clinker. This process generates fine particulate kiln dust, which is captured by air pollution control systems for recycling or disposal (Arulrajah et al., 2017; Barnat-Hunek et al., 2018). Kiln dust accounts

for 2–20 % of the kiln output, depending on process conditions and gas flow rates (Al-Refeai and Al-Karni, 1999; Elbaz et al., 2019; Al-Bakri et al., 2022; Ahmed et al., 2023). Lime kiln dust (LKD) mainly consists of unreacted limestone ($CaCO_3$) and calcium (hydr)oxides (CaO or $Ca(OH)_2$), while cement kiln dust (CKD) is a mix of unreacted feedstock, clinker dust, fuel ash, halides, and other volatiles (Sreekrishnavilasam et al., 2006; Ban et al., 2022). Both types of kiln dust contain a substantial fraction of CaO, and $Ca(OH)_2$, which hence provides alkalinization potential upon addition to seawater (Eq. (3)).

Hence, given their large-scale production, LKD and CKD show a potential as feedstocks for OAE, which is investigated and quantified here through laboratory experiments. OAE suitability was assessed by: (1) evaluating dissolution kinetics and alkalinity generation in seawater, and (2) monitoring changes in seawater properties during dissolution, specifically turbidity, trace elements, and pH increases. These effects were compared to existing environmental guidelines to assess potential risks to marine ecosystems.

## 2 Material and methods

### 2.1 Solid phase characterization

The CKD and LKD were oven-dried at 40°C for 72 h before experimental use. Grain size distribution was determined with a Malvern Mastersizer 3000 equipped with a Hydro LV dispersion system and operated at a stirring speed of 3000 RPM and no sonication prior to measurement. The dry solid phase density of the kiln dust was determined by measuring water displacement in a graduated cylinder (Dan-Asabe et al., 2013). To characterize the elemental composition, a 125 mg aliquot of dry, ball-milled (<2 µm) kiln dust was added to Teflon vessels and digested overnight at 90°C in a heat block with a mixture of 1.5 mL $HClO_4$, 1 mL $HNO_3$, and 2.5 mL HF. After cooling, the caps were removed, and the vessels were heated to 140°C to evaporate HF, leaving a gel-like residue. Next, 25 mL of 4.5 % $HNO_3$ was added to each vessel, which was then capped and heated at 90°C for 2 hours. The resulting solutions were diluted and analyzed for elemental composition using ICP-OES (Avio 500, Perkin-Elmer) at the GeoLab (Utrecht University, The Netherlands). Quality control measures included a blank, two certified river clay standards (ISE 921), and a duplicate sample. Recorded elemental concentration were between 98 and 109 % of certified values. The mineralogical composition of the samples was determined in duplicate via quantitative X-ray diffraction (XRD) on a Bruker D8 Advance Eco diffractometer (Cu Kα, 40 kV, 25 mA) over 5–70° 2θ with 0.015° steps and 0.5 s per

step. Samples were rotated at 10 rpm using a 10 mm variable divergence slit, and patterns were recorded with a LynxEye XE-T detector. Phase identification and quantification were performed using EVA and TOPAS (Bruker, V7). The BET specific surface area was determined by $N_2$ adsorption using a Quantachrome NOVA 2200E at the laboratory for Process Engineering for Sustainable Systems (KU Leuven, Belgium). The geometric specific surface area was calculated from the grain size distribution results (Appendix A Sect. A1). Finally, the grain morphology and presence of secondary minerals on kiln dust grains recovered from the dissolution experiments were analyzed via Scanning Electron Microscopy (SEM) using a Phenom ProX SEM equipped with an energy dispersive spectrometer (EDS), operated at an accelerating voltage of 15 kV. For SEM analysis, the dried kiln dust was mounted on aluminum (Al) pin stubs using double-sided carbon tape.

## 2.2 Dissolution experiments

The dissolution kinetics of LKD and CKD in seawater were assessed in two separate experiments (Table 1), hereafter referred to as experiments I and II. In experiment I, the short-term dissolution behaviour was tracked via continuous pH monitoring. Filtered (<0.2 µm) seawater from the Eastern Scheldt (saline water body in The Netherlands adjacent to the North Sea; salinity $32.3 \pm 0.5$) was obtained from Stichting Zeeschelp (Kamperland, The Netherlands). The filtered seawater (FSW) was aerated for 24 hours before use, yielding an initial seawater $pCO_2$ of 458–557 µatm. The initial seawater pH on a total scale ($pH_T$) and total alkalinity ($A_T$) were analysed (as described in Sect. 2.3). For $A_T$ analysis, 55 mL seawater was collected in duplicate using a 60 mL plastic syringe, and filtered (0.8/0.2 µm polyethersulfone (PES) membrane). Based on preliminary tests, three concentrations of CKD (30, 130, and 309 mg kg$^{-1}$) and LKD (11, 48, and 113 mg kg$^{-1}$) were selected to target specific aragonite saturation states ($\Omega_{Arg} = 3.6, 5.7$ and $9.7$) (Table 1). Experiment I was carried out in triplicate. Kiln dusts were weighed in small aluminium (Al) foil cups using a micro balance (XP26 Excellence Plus, Mettler Toledo) and then transferred to 200 mL polystyrene vials with polyethylene screw caps containing approximately 200 mL of FSW. These small-scale laboratory experiments provide a high-throughput, cost-effective first assessment of a material's suitability for OAE. Vials were weighed on an analytical balance (Sartorius TE3102S) to precisely determine the mass of added seawater. Plastic vials were rinsed with 0.5 M HCl and ultrapure water (PURELAB® flex 3, Elga Veolia) before usage. The pH electrode was inserted through a hole in the vial cap, which fit tightly to minimize atmospheric $CO_2$ exchange. Vials were wrapped in tape to block light and contained minimal headspace. Subsequently, the $pH_T$ of the suspension was measured every minute over a period 8 hours (see

pH$_T$ measurement procedure below). Seawater temperature was kept constant at 20°C during the incubation by means of a water bath (T100, Grant). Magnetic stirring was applied at a rate of 700 rotation per minute (RPM) to ensure good mixing of the suspension and to create optimal dissolution conditions. At the end of experiment I (after ~8 hours), vials were opened and 55 mL of seawater was collected in duplicate for A$_T$ analysis. Salinity was measured, and the samples were subsequently analysed for A$_T$ (see Sect. 2.3).

To assess the alkalinity generation potential and the possibility of secondary mineral formation, we conducted a second dissolution experiment with incubation periods of one and 15 days. The one-day (i.e. 24 h) incubation ensured complete dissolution of the reactive phases in the kiln dusts, while the 15-day incubation allowed for the verification of secondary mineral precipitation, in case this would occur. At the start of experiment II, clean 200 mL plastic vials were filled with 200 mL of FSW on an analytical balance (TE3102S, Sartorius). Different amounts of LKD (11–111 mg kg$^{-1}$) and CKD (30–308 mg kg$^{-1}$) were added to achieve a target $\Omega_{Arg}$ ranging from 3.6 to 9.7 (Table 1). A control containing only 200 mL of FSW without kiln dust was also included. Vials were closed tightly and had minimal headspace to minimize gas exchange with the atmosphere. Experiment II was conducted in triplicate at ambient room temperature (17.5–22.7 °C). Vials were subsequently incubated on bottle rollers (ThermoFisher Scientific) for 1 or 15 days at 14 RPM, a speed sufficient to keep particles suspended and ensure optimal dissolution conditions. Duplicate samples for dissolved inorganic carbon (DIC), dissolved metals, turbidity, and A$_T$ analysis were collected on both sampling days by drawing water with a syringe right after opening the vials (analytical procedures are described in Sect. 2.3). DIC samples were collected first to minimize the exposure time to the atmosphere. Nevertheless, some CO$_2$ exchange inevitably occurred during the incubations because the vials contained a small headspace, meaning the solutions could partially re-equilibrate with the air. However, this also reflects natural deployment conditions, where atmospheric CO$_2$ exchange occurs alongside alkalinization, although at a slower rate. Samples for A$_T$, DIC, and dissolved metals were filtered using 0.8/0.2 µm PES membrane filters. DIC samples (12 mL) were fixed with 10 µL of saturated HgCl$_2$ solution and stored at 4°C in 12 mL exetainers until analysis. Dissolved metal samples were acidified with TraceMetal™ Grade 67–69 % nitric acid to a final acid concentration of 1.4 % (v/v) and stored at -20°C prior until analysis. The remaining suspension in the incubation vials was filtered through a 0.2 µm polycarbonate membrane filter to collect solids, which were rinsed with deionized water and then oven dried at 40°C in preparation for SEM-EDX analysis (see Sect. 2.1).

**Table 1.** Experimental specifications of the two dissolution experiments (I and II). Target aragonite saturation state ($\Omega_{Arg}$), concentrations of lime kiln dust (LKD), cement kiln dust (CKD), incubation time and temperature, and seawater stirring method are provided.

| Exp. | Target $\Omega_{Arg}$ | LKD concentration (mg kg⁻¹) | CKD concentration (mg kg⁻¹) | Incubation time (days) | Temperature (°C) | Stirring method |
|---|---|---|---|---|---|---|
| I | 3.6 | 10.9 | 29.9 | 0.33 | 20 | Magnetic (700 RPM) |
|  | 5.7 | 47.7 | 130.3 |  |  |  |
|  | 9.7 | 113.07 | 309.2 |  |  |  |
| II | 3.0 | 0 | 0 | 1 or 15 | 17.5–22.7 | Rotation (14 RPM) |
|  | 3.6 | 10.75 | 30.12 |  |  |  |
|  | 4.2 | 20.96 | 47.93 |  |  |  |
|  | 5.2 | 37.15 | 88.85 |  |  |  |
|  | 5.7 | 46.49 | 129.12 |  |  |  |
|  | 7 | 69.12 | 191.81 |  |  |  |
|  | 8.4 | 90.94 | 253.35 |  |  |  |
|  | 9.7 | 110.70 | 307.84 |  |  |  |

130

## 2.3  Seawater physicochemical analyses

Seawater salinity and pH$_T$ were measured with an Orion™ DuraProbe™ 4-Cell conductivity probe (Thermo Scientific) and Unitrode pH electrode (Metrohm) connected to an Orion Star A215 pH/conductivity meter. The pH electrode was calibrated

using 35 salinity TRIS (2-amino-2-hydroxy-1,3-propanediol) and AMP (2-aminopyridine) buffers, and the $pH_T$ was calculated following Dickson et al. (2007).

For $A_T$ analysis, samples were titrated with 0.1 M HCl using an automated titrator setup (888 Titrando with 814 USB Sample Processor, Metrohm). The titrant was calibrated with certified reference material (batch 209; OCADS). $A_T$ was derived from the titrant volume and electromotive force measurements recorded by the Unitrode pH electrode, using a non-linear least-squares method as described by Dickson et al. (2007). Blank FSW samples were analyzed at the start of the run, after every tenth sample, and at the end for quality control, yielding relative standard deviations (RSD) smaller than 1.1 %. The specific $A_T$ release $\Delta N_{AT}(t)$ at a given incubation time $t$, and expressed in mmol $A_T$ per g of kiln dust, was derived via

$$\Delta N_{AT}(t) = (A_T(t) - A_T(t_0)) \frac{m_{FSW}}{m_{KD}} \tag{4}$$

Here, $A_T(t)$ represents the seawater alkalinity (mmol kg$^{-1}$) at a given incubation time, $A_T(t_0)$ is the alkalinity (mmol kg$^{-1}$) of the blank FSW at the start of the experiment, $m_{FSW}$ is the mass of FSW added to the incubation vials (kg), and $m_{KD}$ is the mass of added kiln dust (g).

In experiment I, the $A_T$ was only measured at the begin and end of the experiment, and so only one $\Delta N_{AT}(t)$ value can be calculated. To verify the observed alkalinity release, we calculated a theoretical value based on measured $pH_T$. To this end, the initial DIC of the seawater was first calculated from measured values of $pH_T$, $A_T$, temperature, and salinity using the AquaEnv package in R with default settings (Hofmann et al., 2010). Under the assumption that DIC remained constant throughout the experiment, the corresponding $A_T$ was then calculated from the measured $pH_T$ and fixed DIC. The difference between this calculated $A_T$ and the starting $A_T$ was used to compute $\Delta N_{AT}(t)$, and the value at the end (8 hours) was compared to the measured $\Delta N_{AT}(t)$.

For experiment I, the fraction of reactive phases in the kiln dust that dissolved over time, $\chi_{diss}(t)$ (%), was determined by normalizing $\Delta N_{AT}(t)$ to the maximum experimentally observed specific $A_T$ increase ($\Delta N_{AT,max}$):

$$\chi_{diss}(t) = \frac{\Delta N_{AT}(t)}{\Delta N_{AT,max}} 100 \tag{5}$$

Seawater DIC concentrations were measured using a DIC analyzer (AS-C6L, Apollo SciTech) coupled to a trace gas analyzer (LI-7815, LI-COR). Measurements were repeated until the relative standard deviation (RSD) for at least three repeats was ≤0.1 %. DIC concentrations were determined using a calibration curve based on an internal standard solution (0.002 M NaHCO$_3$)

adjusted to a salinity of 30 with NaCl and spiked with 0.05 % (v/v) saturated $HgCl_2$. This internal standard was calibrated

against certified reference material (CRM; batch 209, OCADS). For quality control, CRM (batch 209, OCADS) was analyzed

at the start and end of the sequence, and the internal standard was run at the start and after every eight samples. Quality control

checks consistently yielded an RSD ≤0.25 %.

The remaining seawater carbonate chemistry parameters, including the aragonite and calcite saturation state and seawater $pH_T$,

were calculated using the AquaEnv package in R (Hofmann et al., 2010). Measured $A_T$, DIC, salinity, and temperature were

given as input values, with all other parameters set to their default values.

Seawater samples for dissolved trace metal analysis were thawed and diluted 20-fold with 2 % (v/v) TraceSELECT™ $HNO_3$

(Honeywell Fluka) to a final volume of 10 mL. Before analysis, samples were spiked with 100 µL of an internal standard

solution (10 ppm Y; Alfa Aesar) and analyzed using high-resolution inductively coupled plasma mass spectrometry (HR-ICP-

MS; Agilent 7850) at the ELCAT group, University of Antwerp, Belgium.

Seawater turbidity was measured with a HI98713 ISO portable turbidity meter (Hanna Instruments) which was calibrated with

four turbidity standards (<0.1, 15, 100, and 750 FNU, Hanna Instruments) before each use.

## 2.4 Saturation state calculations

Saturation index (SI) values for kiln dust mineral phases were calculated using PHREEQC Interactive (version 3.7.3-15968)

with the LLNL thermodynamic database (Parkhurst and Appelo, 2013). Saturation indices were converted to saturation states

($\Omega$) according to $\Omega = 10^{SI}$. Input parameters included measured seawater temperature, salinity, $A_T$, and $pH_T$. Major constituent

concentrations (Cl, Na, Mg, K, Ca, $SO_4$) were derived based on the average composition of natural seawater, scaled to

measured salinity (Hem, 1985). Aragonite and calcite saturation states were not computed in PHREEQC but were instead

calculated using the AquaEnv package in R, as described previously (Hofmann et al., 2010). AquaEnv uses carbonic acid

dissociation constants from Lueker et al. (2000) and solubility product constants for $CaCO_3$ from Mucci (1983), which differ

from the thermodynamic data used in the LLNL PHREEQC database to describe the carbonate system. We used the AquaEnv

approach to remain consistent with the methodology commonly applied in most OAE studies.

## 2.5 Statistical analyses

Differences in seawater physico-chemistry across kiln dust concentrations and incubation times were assessed using two-way analysis of variance (ANOVA). The best fitting models were determined by the ANOVA and the lowest Akaike Information Criterion (AIC). Normality and homoscedasticity of residuals were evaluated both visually (via QQ and residual plots) and statistically (via Shapiro-Wilk and Levene's tests). Post-hoc pairwise comparisons were performed using estimated marginal means (EMMs) with Holm-adjusted p-values. Comparisons were conducted within each concentration, adjusting for incubation time, and vice versa. Data are presented as mean ± standard deviation (S.D.), unless otherwise specified. All statistical analyses were performed in RStudio (version 2024.12.0+467) using R version 4.3.3 (R Core Team, 2022).

## 3   Results

### 3.1 Kiln dust physicochemical properties

XRD analysis revealed that the kiln dusts contained substantial amounts of calcite ($CaCO_3$; CKD ~37 % and LKD ~54 %; Table 2) and amorphous phases (~34 % in CKD and ~14 % in LKD). Furthermore, lime (CaO), portlandite ($Ca(OH)_2$), quartz ($SiO_2$) and anhydrite ($CaSO_4$) were additionally present in both kiln dusts at concentrations of 0.1–20 %. Finally, CKD also contained 0.9 % sylvite (KCl), 9.5 % syngenite ($K_2Ca(SO_4)_2.H_2O$), and 3.9 % Aphthitalite ($(K,Na)_3Na(SO_4)_2$) (Table 2). Furthermore, minor phases including 2.1% larnite/$\beta$-$C_2S$ ($Ca_2SiO_4$), 0.68% hematite ($Fe_2O_3$), and 0.56% maghemite ($\gamma$-$Fe_2O_3$) were identified with low confidence in one of the duplicate CKD samples analyzed by XRD. Both materials exhibited a relatively wide grain size distribution (Table 2 and Appendix A Fig. A1), but CKD (D50 = 8.4 ± 0.1 µm) was significantly finer than LKD (D50 = 72 ± 4 µm). The observed elemental composition was generally in line with the XRD results, showing high calcium contents in both CKD (27.8 wt%) and LKD (44.9 wt%), which fall within the range previously reported for CKD (14–46 %) and LKD (20–49 wt%) (Collins and Emery, 1983; Pavía and Regan, 2010; Latif et al., 2015; Drapanauskaite et al., 2021; Dvorkin and Zhitkovsky, 2023). While LKD exhibited relatively low levels of trace elements, CKD showed elevated concentrations, particularly of Zn (0.65 wt%) and Pb (0.15 wt%) (Table 2). Further details on the elemental composition are provided in Appendix A Table A1.

Based on the mineralogical composition, the theoretical alkalinity release upon dissolution in seawater was calculated from the reaction stoichiometry. Saturation state analysis (Appendix B Fig. B2) show that seawater is undersaturated ($\Omega < 1$) with respect to most mineral phases present in the kiln dusts, with the exception of calcite and quartz. This indicates that dissolution is thermodynamically favourable, although it may be limited by kinetic constraints. Among the undersaturated minerals, anhydrite, sylvite, aphthitalite, and syngenite do not contribute to alkalinity upon dissolution in seawater. Furthermore, the

hematite and maghemite present in the CKD are essentially insoluble under oxic conditions in natural seawater, and filtration would remove Fe-reducing bacteria capable of enhancing their dissolution (Canfield, 1989). In contrast, the dissolution of portlandite and lime each produces two moles of $A_T$ per mole (Eq. 3), while larnite (potentially present in the CKD) would yield four moles of $A_T$ per mole upon dissolution (Brand et al., 2019). Overall, complete dissolution of these phases corresponds to maximum alkalinity contributions of 8.8 mmol $g^{-1}$ for the LKD and 1.7 mmol $g^{-1}$ for the CKD, respectively.

**Table 2. Physicochemical properties of the experimental cement kiln dust (CKD) and lime kiln dust (LKD). ND indicates that the phases were not detectable. The minor phases larnite/beta-calcium disilicate ($\beta$-C$_2$S; 2.1%), hematite (Fe$_2$O$_3$; 0.68%), and maghemite ($\gamma$-Fe$_2$O$_3$; 0.56%) were detected with low confidence in one of the duplicate samples and are therefore mentioned here but not included in the table. The complete measured elemental composition is provided in Appendix A Table A1.**

| Mineral composition (% w/w) | CKD | LKD |
|---|---|---|
| Calcite (CaCO$_3$) | 37.0 | 53.8 |
| Lime (CaO) | 1.5 | 9.4 |
| Portlandite (Ca(OH)$_2$) | 2.8 | 20.1 |
| Quartz (SiO$_2$) | 3.4 | 0.13 |
| Anhydrite (CaSO$_4$) | 4.7 | 2.2 |
| Sylvite (KCl) | 0.9 | ND |
| Syngenite (K$_2$Ca(SO$_4$)$_2$.H$_2$O) | 9.5 | ND |
| Aphthitalite ((K,Na)$_3$Na(SO$_4$)$_2$) | 3.9 | ND |
| Amorphous phases | 34.3 | 14.4 |
| **Theoretical alkalinization potential (mmol g$^{-1}$)** | 1.7 | 8.8 |
| **Textural composition** | | |
| D10 ($\mu$m) | 2.2 ± 0.02 | 6.4 ± 0.2 |
| D25 ($\mu$m) | 3.9 ± 0.02 | 22 ± 1 |
| D50 ($\mu$m) | 8.4 ± 0.05 | 72 ± 4 |

| | | |
|---|---|---|
| D75 (µm) | 24 ± 0.3 | 172 ± 12 |
| D90 (µm) | 49 ± 0.9 | 327 ± 35 |
| Geometric specific surface area (m² g⁻¹) | 0.455 ± 0.007 | 0.121 ± 0.004 |
| BET specific surface area (m² g⁻¹) | 4.2 | 3.2 |
| Particle density (g cm⁻³) | 2.71 ± 0.18 | 2.87 ± 0.14 |
| **Elemental composition (% w/w)** | | |
| Ca | 27.8 | 44.9 |
| Mg | 0.56 | 0.22 |
| Zn | 0.65 | 0.0029 |
| Pb | 0.15 | <0.000014 |

## 3.2 Kiln dust dissolution speed

The dissolution kinetics of CKD and LKD in seawater (salinity 32 ± 0.1, temperature 20°C, initial $pH_T$ of 8.05 ± 0.03) were investigated in experiment I over a period of 8 hours at three different kiln dust concentrations under continuous stirring. A rapid increase in seawater $pH_T$ was observed for both kiln dusts: about 69 ± 8 % of the total $pH_T$ rise occurred within the first ~10 minutes, after which the $pH_T$ continued to slowly increase for the rest of the 8-hour incubation (Fig. 1a). The final $pH_T$ increase $\Delta pH_T = pH_T(t_{end}) - pH_T(t_0)$ was greater for LKD than for CKD and increased with higher kiln dust concentrations (Fig. 1c).

Upon dissolution, the percentage of reactive phases that had effectively dissolved $\chi_{diss}(t)$ rapidly increased with time, with 50 % of the reactive phases dissolved within 9.5 ± 3 minutes for LKD and within 9.9 ± 2.4 minutes for CKD (Fig. 1b). At the lowest kiln dust concentration, approximately 83 ± 9 % of the reactive phases in LKD and 72 ± 12 % in CKD had dissolved after one hour, which increased to 88 ± 7 % and 82 ± 17 % respectively after 8 hours. At higher kiln dust concentrations, the percentage of dissolved reactive phases was generally lower (Fig. 1b). This effect is likely caused by secondary mineral precipitation, and not so much by reduced dissolution, as further discussed in Sect. 3.3.

Kiln dust dissolution in seawater resulted in a concomitant increase in alkalinity, $\Delta A_T = A_T(t) - A_T(t_0)$, which increased with higher kiln dust concentrations (Fig. 1d), and attained values ranging from 60 to 447 µmol kg⁻¹ for CKD and 110 to 416 µmol

235  kg$^{-1}$ for LKD, depending on the applied kiln dust concentration. Notably, the relationship between $\Delta A_T$ and kiln dust addition was non-linear and showed a saturating effect (Fig. 1d), suggesting a decreased specific alkalinity release at higher kiln dust concentrations.

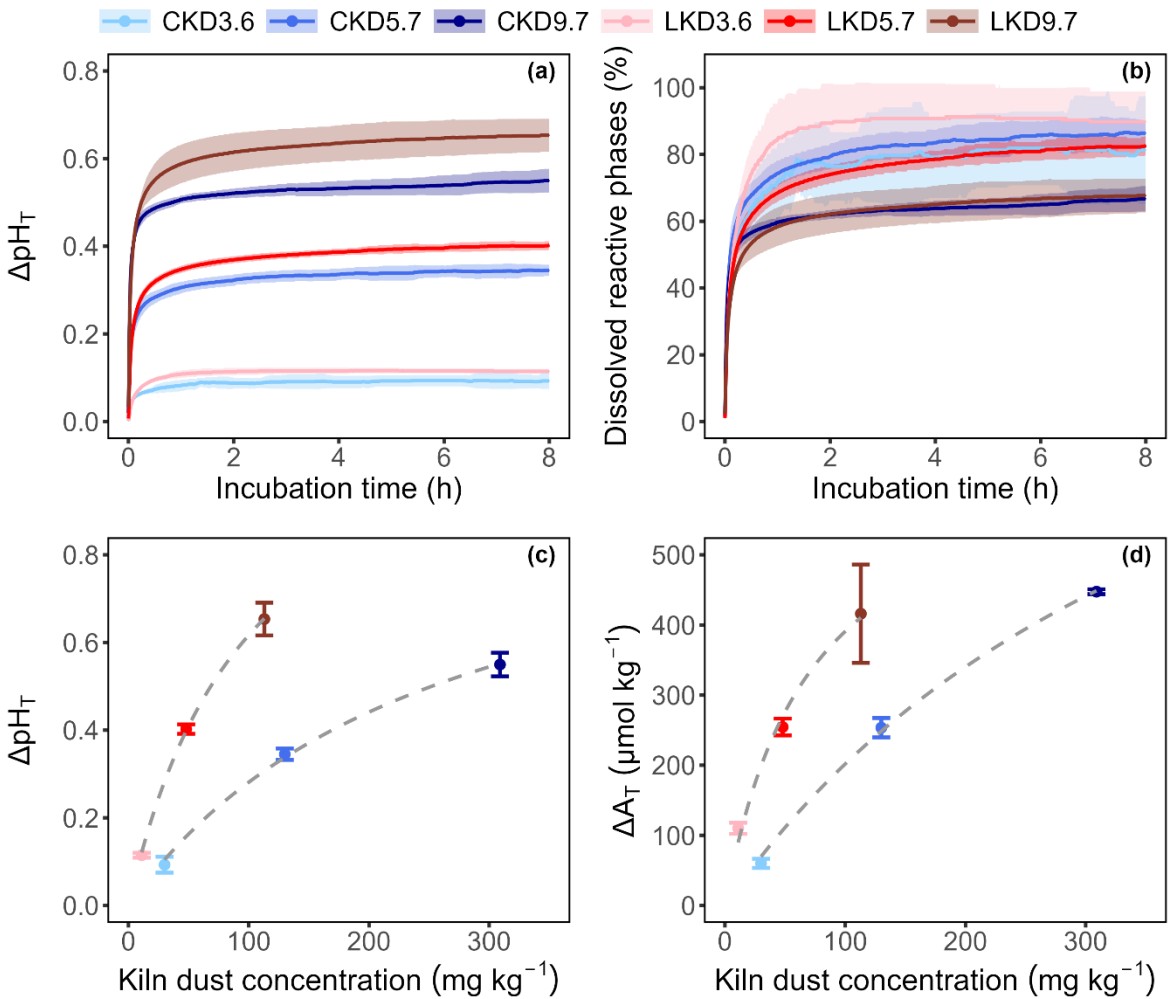

**Figure 1. Results (Mean ± S.D., N =3) obtained in experiment I for three concentrations of cement kiln dust (CKD, in blue) and lime**
**kiln dust (LKD, in red). (a) Change in seawater pH$_T$ ($\Delta$pH$_T$) as a function of incubation time. (b) Estimated percentage of reactive kiln dust phases ($\chi_{diss}$) dissolved as a function of incubation time. Change in (c) seawater pH$_T$ and (d) total alkalinity ($\Delta A_T$) after 8 hours as a function of kiln dust concentration. Fitted non-linear least squares curves are shown as a grey dashed line in (c) and (d).**

### 3.3  Alkalinity generation

In experiment II, we investigated the alkalinity release from LKD and CKD after 1 and 15 days of incubation across a range

of kiln dust concentrations. Dissolution resulted in a change in seawater pH$_T$ and A$_T$ that markedly varied with kiln dust

concentration (Fig. 2a and 2b). After one day, the $pH_T$ showed a non-linear (saturating) increase with the kiln dust concentrations for both LKD and CKD (Fig. 2a), consistent with the results of experiment I after 8 hours (Fig. 1c). Yet after 15 days of incubation, the $\Delta pH_T$ curve showed a marked difference between LKD and CKD. For CKD, the $\Delta pH_T$ curve at 15 days showed a similar saturating shape as after 1 day, though with slightly decreased $\Delta pH_T$ values at higher kiln dust concentrations (suggesting a process at play that reduced $pH_T$). For LKD, the $\Delta pH_T$ curve at 15 days was entirely different compared to day 1, with a marked decrease in $\Delta pH_T$ at higher concentrations. For the highest LKD concentration examined (111 mg kg$^{-1}$), the $pH_T$ after 15 days was almost the same as at the start ($\Delta pH_T = 0.17 \pm 0.14$) (Fig. 2a).

The observed changes in alkalinity (Fig. 2b) were congruent with those seen for $\Delta pH_T$. After one day, $\Delta A_T$ showed a monotonous increase with the CKD concentration, while for LKD, the $\Delta A_T$ curve reached a maximum at 69 mg kg$^{-1}$ and decreased at higher application concentrations (Fig. 2b). The $\Delta A_T$ curves for LKD and CKD obtained at 15 days of incubation were markedly different from those at day 1, showing reduced $\Delta A_T$ values at higher kiln dust concentrations, indicative of a process that consumes alkalinity. Notable, for LKD, $\Delta A_T$ became negative at higher concentrations (Fig. 2b), indicating a removal of alkalinity compared to the initial situation.

The specific alkalinity release quantifies the alkalinity release per mass of kiln dust added, and generally decreased for higher concentrations (Fig. 2c). The highest values for the specific alkalinity release were obtained at the lowest kiln dust concentrations applied (21 mg kg$^{-1}$ and lower for LKD, 89 mg kg$^{-1}$ or lower for CKD). The maximum specific alkalinity release for LKD ($8.02 \pm 0.53$ mmol g$^{-1}$) was more than three times higher than that of CKD ($2.38 \pm 0.16$ mmol g$^{-1}$). Moreover, the LKD value was in good agreement with the theoretical prediction (8.8 mmol g$^{-1}$; see section 3.1), while the CKD value deviated more substantially from the theoretical estimate (1.7 mmol g$^{-1}$). The specific alkalinity release after 15 days showed a clear difference with day 1, which was more pronounced for LKD compared to CKD. At the low application concentrations, the specific $A_T$ release after 15 days of incubation was not statistically significantly different from the release after 1 day (Fig. 2c), thus indicating that all $A_T$-generating reactive phases had dissolved within the first day and no alkalinity removal took place. Yet, the specific $A_T$ release decreased substantially at higher kiln dust concentrations, with a more pronounced decline for LKD compared to CKD. In the case of LKD, the specific $A_T$ release became even negative after 15 days of incubation, thus indicating overall alkalinity consumption rather than production (Fig. 2c).

To verify whether secondary carbonate precipitation could be responsible for the observed alkalinity consumption, we calculated the seawater aragonite saturation state ($\Omega_{Arg}$). The increase in seawater total alkalinity ($A_T$) from kiln dust dissolution led to a corresponding rise in aragonite saturation state ($\Omega_{Arg}$) from an initial value of $3.0 \pm 0.1$ at the start of the experiment II to $9.3 \pm 0.3$ and $10.3 \pm 0.1$ after 1 day of incubation at the highest concentrations of LKD and CKD, respectively (Fig. 2d).

Moreover, a significant decrease in $\Omega_{Arg}$ was observed at day 15 compared to day 1 for LKD concentrations above 21 mg kg$^{-1}$ and for CKD concentrations above 89 mg kg$^{-1}$, suggesting alkalinity scavenging by secondary mineral precipitation. This decrease became more pronounced at higher application concentrations and was notably greater for LKD compared to CKD. For the highest LKD concentration investigated (111 mg kg$^{-1}$), $\Omega_{Arg}$ after 15 days ($2.5 \pm 0.6$) was not significantly different from the initial $\Omega_{Arg}$ value (Fig. 2d).

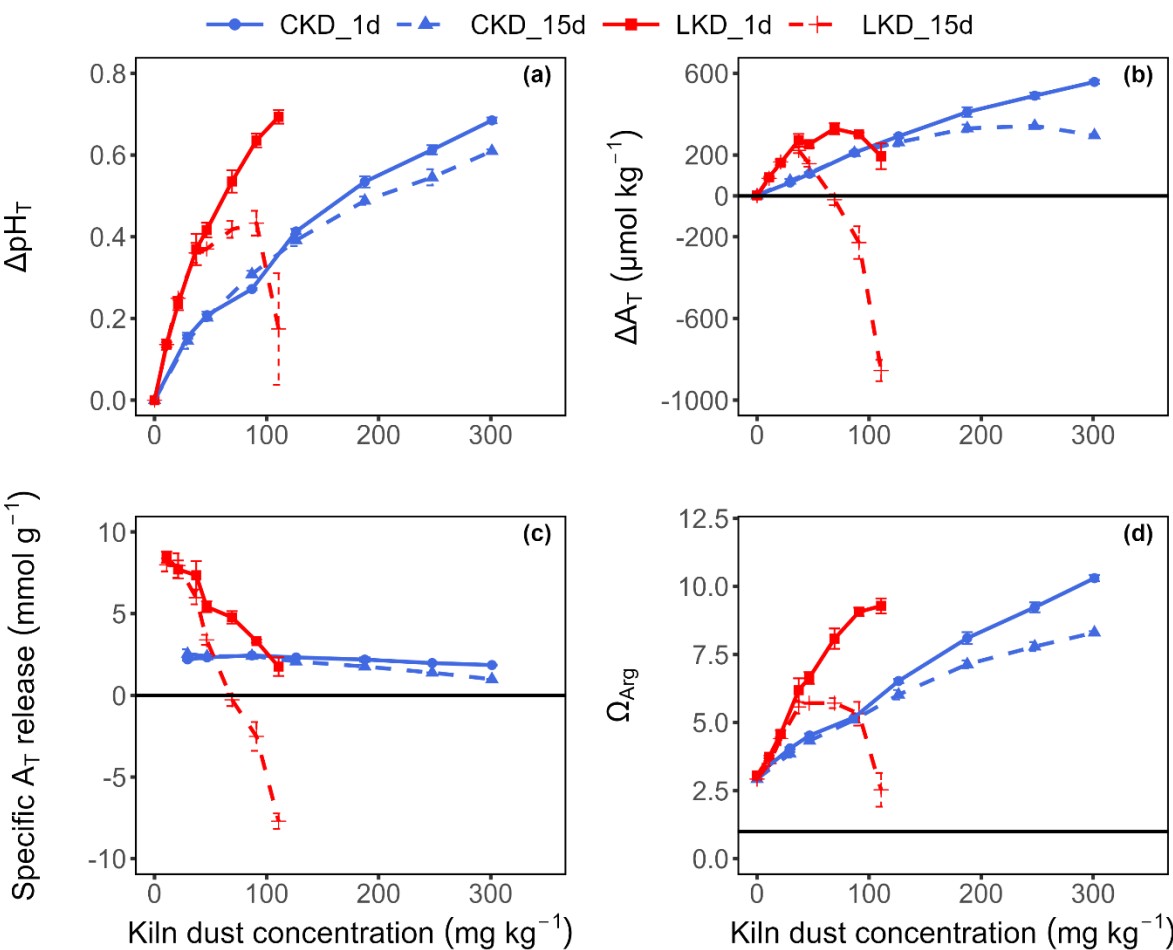

**Figure 2. Results obtained in experiment II for different concentrations of cement kiln dust (CKD, blue) and lime kiln dust (LKD, red). (a) Change in seawater $pH_T$ ($\Delta pH_T$), (b) total alkalinity ($\Delta A_T$, expressed in µmol kg$^{-1}$), (c) Specific $A_T$ release (mmol mg$^{-1}$) and (d) aragonite saturation state ($\Omega_{Arg}$) as a function of the kiln dust application concentration (mg kg$^{-1}$). Results (as mean ± S.D., N=3) after 1 day (solid lines) and 15 days of incubation (dashed lines) are shown.**

## 3.4 Mineral morphology and secondary mineral precipitation

The morphology of kiln dust particles was compared before and after chemical weathering (i.e., fresh material versus samples retrieved after 15 days in experiment II). Fresh lime kiln dust (LKD) consisted of a heterogeneous mixture of particles of varying size and shape, dominated by small (<10 µm), irregular, calcium-rich particles (~10–20 at%) (Fig. 3a). In weathered LKD, the abundance of these fine particles decreased (Fig. 3c, e), while larger particles developed rough surface texture (representative particle indicated by red arrow in Fig. 3c), suggesting dissolution of surface phases. After 15 days of incubation at the highest LKD concentration, most particles were extensively coated with bundles of needle-like, calcium-rich precipitates (Fig. 3e).

Fresh cement kiln dust (CKD) also contained irregularly shaped particles of various sizes (Fig. 3b). In addition, spherical fly ash particles of varying diameters were prominent in both fresh and weathered CKD samples and showed no signs of weathering during experiment II (red arrows in Fig. 3b, d, f). In contrast to LKD, weathered CKD particles showed neither roughened surfaces nor secondary calcium carbonate precipitation, regardless of concentration or incubation duration (Fig. 3d, f).

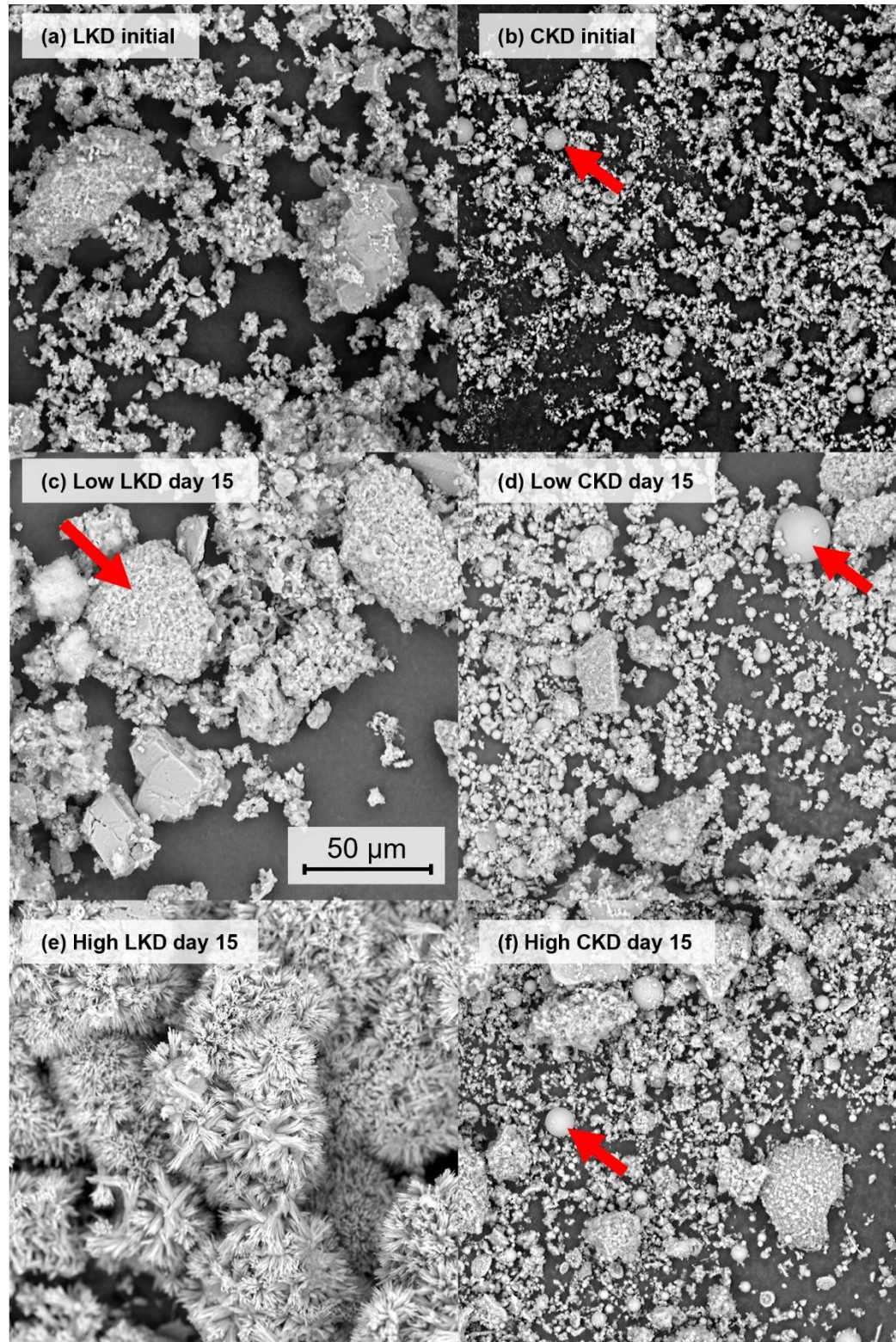

**Figure 3: Representative scanning electron microscopy (SEM) images of kiln dust particles at different time points in experiment II. (a) Fresh lime kiln dust (LKD) and (b) fresh cement kiln dust (CKD). Images (c) and (d) show CKD and LKD particles after 15 days of incubation at low application concentrations (88.85 mg kg⁻¹ and 20.96 mg kg⁻¹, respectively). Images (e) and (f) show CKD and LKD particles incubated for 15 days at the highest application concentrations (307.84 mg kg⁻¹ and 110.7 mg kg⁻¹, respectively). Red arrows in panels (b), (d), and (f) mark fly ash particles, while the red arrow in panel (c) highlights the rough surface texture of a weathered large LKD particle.**

## 3.5 Turbidity and trace metals

The turbidity in solution after 1 and 15 days increased linearly with the applied kiln dust concentration, as expected. At equivalent application concentrations, turbidity was ~2.4 times higher for CKD than for LKD, consistent with the finer grain size of the CKD (Fig. 4a). After 15 days of incubation, seawater turbidity rose from $0.33 \pm 0.10$ FNU in the initial filtered seawater to $281 \pm 3$ FNU at the highest CKD concentration and $52 \pm 3$ FNU at the highest LKD concentration. Turbidity was slightly but statistically significantly greater on day 15 compared to day 1 at higher kiln dust concentrations (>253 mg kg⁻¹ for CKD; >69 mg kg⁻¹ LKD), possibly due to fragmentation of unreacted kiln dust particles and/or the formation of secondary calcium carbonate providing fine particles in solution (Fig. 4a).

The accumulation of dissolved Fe, Ni, Cu, Zn, As, and Pb was low in both the LKD and CKD incubations and showed no clear dependence on kiln dust concentration (Appendix B Fig. B3). So trace metal release was limited, apart from vanadium (V) in the CKD incubation, which linearly scaled with the kiln dust concentration (Fig. 4b). After 15 days, V accumulation reached $0.51 \pm 0.09$ µmol kg⁻¹ at the highest CKD concentration and $0.008 \pm 0.001$ µmol kg⁻¹ at the highest LKD concentration, respectively. These values agreed relatively well with the expected accumulations of 0.63 µmol kg⁻¹ and 0.0033 µmol kg⁻¹, based on the elemental composition and assuming that V belongs to the dissolvable fraction of the kiln dusts (Appendix B Fig. B4). The dissolved V accumulation after 15 days did not differ significantly from that on day 1, suggesting that V is not involved in secondary reactions. For CKD, concentration-dependent accumulation of Al, Cr, and Mn was also observed, though the accumulation represented only a small fraction of what was expected based on complete dissolution of reactive phases: 2–11 % for Al, 24–35 % for Cr, and 0.6–2 % for Mn (Appendix B Fig. B3-4).

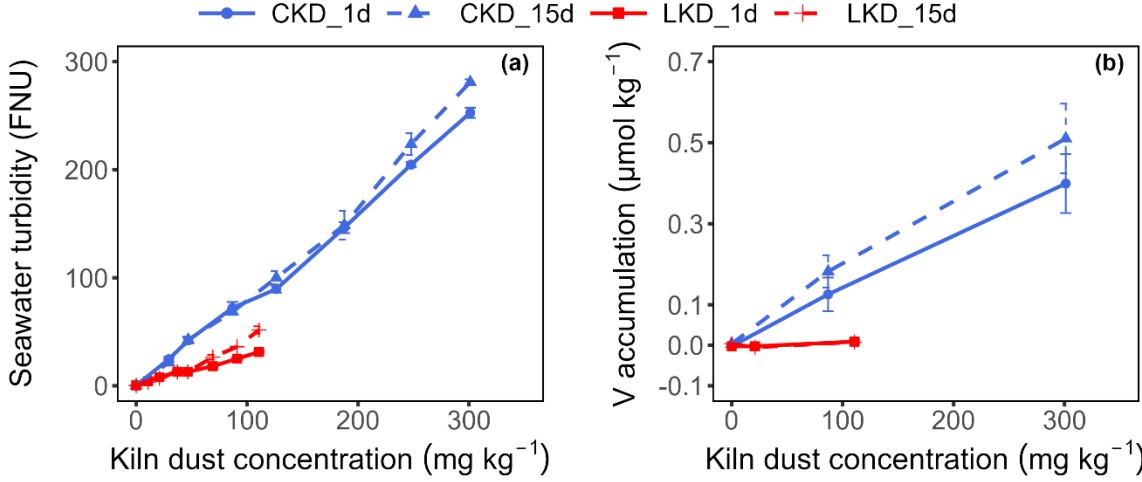

**Figure 4. Impact of dissolution on turbidity and trace metals in experiment II. Results (as mean ± S.D., N=3) are shown for lime kiln dust (LKD, red) and cement kiln dust (CKD, blue) after 1 day (solid lines) or 15 days (dashed lines) of incubation. (a) Seawater turbidity (FNU) and (b) accumulation of dissolved vanadium (V, µmol kg⁻¹) as a function of the kiln dust concentration.**

## 4    Discussion

### 4.1  Dissolution kinetics and alkalinity generation potential

The dissolution kinetics of LKD and CKD were studied in natural seawater (salinity: $32.3 \pm 0.5$; temperature: $17.5–22.7\,°C$) under continuous stirring. LKD mainly consists of calcite ($CaCO_3$), with smaller amounts of quartz, portlandite, lime, anhydrite, mullite, and dolomite (Strydom et al., 1996; Ban et al., 2022). CKD is more compositionally complex, typically containing calcite along with various sulfates, chlorides, silicates, and aluminates, including belite, aphthitalite, spurite, ettringite, arcanite, and ferrite (Ayman et al., 2004; Siddique and Rajor, 2012; Beltagui et al., 2017; Adekunle, 2024; Lee and Choi, 2024; Nikolov et al., 2025). The compositional complexity of kiln dusts underscores the need for detailed mineralogical and chemical characterization to properly assess the CDR potential and environmental risks in OAE applications.

Both materials generated significant alkalinity upon dissolution in seawater, with maximum values of $8.02 \pm 0.53\ \mu mol\,mg^{-1}$ for LKD and $2.38 \pm 0.16\ \mu mol\,mg^{-1}$ for CKD after 24 hours. This alkalinity originated from a fraction of reactive phases contained within the kiln dust (25 % by mass in LKD; 29 % in CKD), which we estimated based on the mineral components that are unstable in seawater. Consequently, both LKD and CKD contained a substantial amount of unreactive phases that

remained inert over the 15-day duration of our short-term experiments. In preliminary tests, replacement of the overlying seawater after 15 days did not result in further dissolution or alkalinization. The residual fraction in LKD and CKD consisted primarily of calcite, which is supersaturated in surface seawater (Appendix B Fig. B2G), preventing its dissolution (Sulpis et al., 2021).

The experimentally observed alkalinity release from LKD ($8.02 \pm 0.53$ mmol g$^{-1}$) was slightly lower than the theoretical value ($8.8$ mmol g$^{-1}$), whereas CKD released substantially more alkalinity ($2.38 \pm 0.16$ mmol g$^{-1}$) than predicted from its mineralogical composition ($1.7$ mmol g$^{-1}$). In LKD, alkalinity release was fully attributed to the dissolution of portlandite ($Ca(OH)_2$) and lime ($CaO$), whereas in CKD these phases explained only about half ($54 \pm 3$ %) of the observed alkalinity release. Calcium silicates (e.g. larnite $Ca_2SiO_4$) are also alkalinity-generating phases that occur in minor amounts in CKD. They originate form the raw materials used in cement production (e.g. iron ore, clay, or shale) and exhibit a relatively high reactivity in water (Brand et al., 2019; Adekunle, 2024). Dissolution of the larnite present in our CKD sample ($\sim 2.1$ %) could account for $17 \pm 1$% of the observed $A_T$ release, which hence provides a substantial additional contribution. The remaining $\sim$29% of the alkalinity released from CKD likely originated from dissolution of amorphous phases, including (partially dehydrated) clay minerals, reactive amorphous silica, and kiln-derived materials such as fly ash or slag (Khanna, 2010; Pavía and Regan, 2010). ICP-OES analysis revealed 8.0 % Ca, 2.7 % K, 0.7 % Na, 0.9 % S, 1.6% Fe and 3 % Al that were not accounted for by the crystalline phases detected via XRD (Table 2 and Table A1). This suggests the possible presence of amorphous calcium aluminosilicates, alkali sulfates, and poorly crystalline CaO or $Ca(OH)_2$, which may have contributed to the remaining alkalinity upon dissolution in seawater (Hu et al., 2024; Nikolov et al., 2025). There is a possibility that some reactive phases remained sequestered in larger particles and did not react with seawater on the time scale of the experiment. However, this fraction is likely minor, since the experimentally observed alkalinity release was close to, or even exceeded, the maximum theoretically predicted values from the mineralogical composition.

The dissolution of kiln dusts was rapid, with 65–92 % of the reactive phases dissolving within the first hour of incubation under continuous stirring (Fig. 1b). This estimate assumes a constant DIC concentration over time, although values could have increased due to $CO_2$ uptake from the (limited) vial headspace or decreased through secondary aragonite precipitation at higher application concentrations. Therefore, the $\chi_{diss}$ values reported (Fig. 1b) should be considered best estimates based on

available data. Nonetheless, the rapid alkalinity release highlights the potential of kiln dusts for OAE. Using our measured particle size distribution and assuming particle sinking follows Stokes' law, all CKD particles and the majority of LKD particles ($85 \pm 2$ % V/V) will remain in the surface ocean mixed layer (assumed to be 200 m) for at least one hour, thus allowing sufficient time for most reactive phases to dissolve and generate alkalinity (Appendix A Sect. A2). However, ocean turbulence and particle aggregation induced by biological exudates can significantly accelerate particle sinking (Yang and Timmermans, 2024) and affect kiln dust dissolution kinetics . Consequently, further research is needed to quantify kiln dust dissolution rates and settling velocities under a range of scenarios that more closely mimic *in situ* hydrodynamic conditions. This information is essential for the careful selection of application areas, ensuring that dissolution of the reactive phases occurs within the ocean mixed layer.

## 4.2 Emergence and prohibition of secondary precipitation

Alkalinity addition to seawater should avoid triggering secondary precipitation reactions that consume alkalinity, as these reduce the overall efficiency of OAE. Specific $A_T$ release significantly decreased at concentrations above 21 mg kg$^{-1}$ for LKD and 89 mg kg$^{-1}$ for CKD, with greater reductions at higher kiln dust concentrations and longer incubation times (Fig. 2c). Furthermore, the decrease was stronger for LKD relative to CKD at equivalent aragonite saturation states (Fig. 2c and 2d). The observed reduction in specific $A_T$ release can be attributed to secondary mineral precipitation, as indicated by the reduction in seawater DIC concentrations (Appendix B Fig. B1) and the formation of Ca-rich needle-like structures on weathered LKD grains (Fig. 3e). These needles resemble the early developmental stage of aragonite precipitates, as described by Suitner et al. (2024). No significant $A_T$ loss occurred at $\Omega_{Arg}$ values of $5.1 \pm 0.07$ for CKD and $4.5 \pm 0.1$ for LKD, but significant loss was observed at higher kiln dust concentrations. This aligns with the previously documented $\Omega_{Arg} = 5$ threshold for secondary aragonite precipitation, when fine-grained (<63 µm) quick lime (CaO) or slaked lime (Ca(OH)$_2$) powder from a chemical and industrial supplier are added to natural seawater at 35 salinity (Moras et al., 2022). While the $\Omega_{Arg}$ precipitation threshold was similar for LKD and CKD, the CKD treatment showed a lower precipitation rate, possibly because CKD has a lower content of Ca-rich phases (e.g., calcite, lime and portlandite), which can serve as nucleation sites for aragonite precipitation (Pan et al., 2021; Moras et al., 2022; Suitner et al., 2024). Prolonged exceedance of critical saturation thresholds can trigger "runaway

CaCO$_3$ precipitation", leading to a net A$_T$ loss, as seen at the highest LKD concentration after 15 days (Fig. 2b) (Moras et al., 2022). Under natural conditions, freshly precipitated aragonite may redissolve after dilution in the ship's wake, especially when not yet fully crystallized, recovering some of the lost alkalinity due to secondary precipitation (Hartmann et al., 2023). Aragonite precipitates that settle onto the sediment at the deployment site may undergo further metabolic dissolution provided that geochemical conditions are favourable, offsetting the earlier alkalinity loss (see Section 4.4). However, these fine-grained precipitates could also disperse far from the deployment site, thus complicating monitoring, reporting, and verification (MRV) of CDR via kiln-dust-based OAE. So, despite that some secondary aragonite may redissolve, its formation is best minimized to maximize the alkalinization potential. Based on our temporal dissolution data (Fig. 1a-b), it is recommended to adjust the OAE dispensing and deployment procedure in such a way, that dilution to $\Omega_{Arg} < 5$ occurs within minutes as to minimize secondary mineral precipitation.

### 4.3 Potential ecological impacts of kiln dust dissolution

Mineral-based OAE shows promise as a CDR technique, but its effects on seawater carbonate chemistry, turbidity, and trace element concentrations could lead to adverse ecological impacts that need to be mitigated (Bach et al., 2019; Flipkens et al., 2021). For ocean liming, rapid mineral dissolution can cause acute spikes in pH$_T$ and alkalinity right after discharge, raising potential environmental concerns (Caserini et al., 2021; Varliero et al., 2024). In our study, CKD and LKD caused fast, concentration-dependent seawater pH$_T$ increases (Fig. 1a). Model predictions indicate that pH could rise by 1 to 1.5 units for several minutes during ship-based ocean liming (Caserini et al., 2021), which may have an impact on marine life if pH exceeds 9 (ANZECC and ARMCANZ, 2000; Pedersen and Hansen, 2003; Camatti et al., 2024). To avoid temporary exceedances of pH 9, CKD concentrations should stay below 343–502 mg kg$^{-1}$, and LKD below 102–149 mg kg$^{-1}$ under average surface seawater conditions (A$_T$ = 2350 µmol kg$^{-1}$, DIC = 2100 µmol kg$^{-1}$, salinity = 35, temperature = 10–25 °C). Application concentrations must be tailored to local seawater geochemistry at the deployment site to prevent exceeding the pH 9 threshold. Seawater turbidity rose linearly with kiln dust concentration and was greater for CKD than LKD at equivalent doses, consistent with the finer grain size of CKD (Fig. 4a). Increased turbidity could reduce primary production by obstructing light (Cloern, 1987; Köhler et al., 2013), and impair feeding efficiency in marine suspension feeders (e.g. bivalves, sponges, and tunicates)

(Cheung and Shin, 2005; Bell et al., 2015) and visual foragers (e.g. most marine fish and mammals) (Lowe et al., 2015; Lunt and Smee, 2020). Additionally, the sinking rate of organic carbon to the deep sea could be enhanced by the adsorption of organic molecules onto suspended particles (Santinelli et al., 2024), which may affect ecosystem carbon cycling, but also would provide additional CDR. Turbidity guidelines are designed to protect marine life from harmful increases: for example, in Canada, seawater turbidity should not increase by more than 8 NTU over 24 hours in clear water, or 5 NTU at any time in already turbid water (8–50 NTU background) (Singleton, 2021). To stay within these limits, ambient CKD concentrations must remain below 9.7 mg kg$^{-1}$ in clear water and 6.1 mg kg$^{-1}$ in turbid water, while LKD concentrations should stay below 23.7 and 14.8 mg kg$^{-1}$, respectively (Fig. 4b). In real applications, kiln dust will be rapidly mixed into much larger volumes of surface water, meaning that the allowable concentration in the input stream will depend on the discharge rate, the intensity of local turbulence, and kiln dust settling time (which is primarily determined by initial particle size). Accurate numerical modelling to determine suitable discharge rates therefore requires detailed knowledge of both the environmental conditions at the deployment site and the behaviour of kiln dust particles under varying hydrodynamic conditions (Fennel et al., 2023; Yang and Timmermans, 2024). This information is essential to extrapolate small-scale laboratory results to realistic field scenarios and ensure that concentrations remain below guideline levels.

At the turbidity thresholds, the maximum seawater alkalinity enhancement would range from 119 to 190 µmol $A_T$ kg$^{-1}$ for LKD, and only 15 to 23 µmol $A_T$ kg$^{-1}$ for CKD. The corresponding increases in pH$_T$ (assuming $A_T$ = 2350 µmol kg$^{-1}$, $pCO_2$ = 420 µatm, salinity = 35, and temperature = 10–25 °C) are $\Delta pH_T$ = 0.18–0.30 for LKD and $\Delta pH_T$ = 0.002–0.004 for CKD, both remaining well below the threshold of $\Delta pH_T$ = ~0.9 (rise up to pH$_T$ 9). Moreover, the carrying capacity of natural coastal and shelf ecosystems appears to be large enough to execute LKD- and CKD-based OAE within the existing turbidity constraints. For example, the North Sea has a total volume 54.000 km$^3$ and an average residence time of ~1 year (Lee, 1980; Liu et al., 2019). A hypothetical one-time application of LKD-based OAE across the entire North Sea at the maximum level of 119 µmol $A_T$ kg$^{-1}$ for LKD, would require 823 Mton of LKD, which is ~28 times larger than the global annual LKD production rate (~29 Mton yr$^{-1}$).

The dissolution of alkaline minerals can release trace metals into the environment, which may be beneficial or toxic to marine life (Bach et al., 2019; Flipkens et al., 2021). CKD contained notable amounts of Zn and Pb, while LKD had generally low

trace element levels (Appendix A Table A1). Trace metal content in kiln dusts varies with raw materials, fuels, and kiln

operations, and is typically higher and more variable in CKD (Siddique and Rajor, 2012; Nyström et al., 2019). In experiment

II, metal release from LKD was limited, whereas CKD showed concentration-dependent release of V, Cr, Mn, and Fe (Fig.

4b; Appendix B Fig. B3). Regulatory guidelines exist to protect aquatic life from trace metal toxicity. For example, Tulcan et

al. (2021), proposed a seawater V guideline of 0.022 µmol L$^{-1}$, which would require CKD concentrations to remain below

14.1 mg kg$^{-1}$ to avoid exceedance. Higher application rates may be permissible under other guidelines for V, Cr, or Mn.

Residual kiln dust mixing with surface sediments may elevate metal levels, particularly Zn and Pb from CKD. Sediment

Quality Guidelines (SQGs) aim to protect benthic ecosystems (Hübner et al., 2009; Simpson and Batley, 2016). Assuming full

mixing in the top 10 cm of the sediment, up to 1.4 kg CKD or 74.8 kg LKD per m² could be applied to pristine sediments

without exceeding the strictest marine SQG of 30.2 mg kg$^{-1}$ for Pb (Appendix C). Using a kiln dust bulk density of 0.62 g cm$^{-3}$

(Nikolov et al., 2025), this corresponds to an applied layer of approximately 0.2 cm for CKD and 12 cm for LKD. The exact

layer thickness would of course be dependent on the specific kiln dust properties and local grain packing. These estimates

indicate that burial risk for benthic organisms is small for CKD, but could be considerable for LKD, since the deposition of a

cm-thick layer could substantially impact the resident benthic infauna and epifauna. Moreover, applying LKD at this scale is

also not advisable because it may lead to changes in habitat suitability (e.g., grain size, permeability, organic carbon content)

(Speybroeck et al., 2006; Flipkens et al., 2024) and alter geochemical sediment processes (see Sect. 4.4). Overall, these findings

underscore the need for ecotoxicological testing and cautious application of kiln dust to avoid ecological harm.

## 4.4 Longer term fate of unreacted phases

Both LKD and CKD contained a significant amount of unreactive phases (75 and 71 wt%, respectively) that remained inert

over the 15-day experimental time scale. In coastal and shelf environments, this residual material would rapidly settle to the

seafloor. The residual fraction consists primarily of $CaCO_3$ phases (52 % in CKD and 72 % in LKD). When residual $CaCO_3$

or freshly precipitated secondary $CaCO_3$ become mixed into the seabed through local hydrodynamics and bioturbation,

porewater acidification resulting from microbial degradation of organic matter can trigger metabolic $CaCO_3$ dissolution (Rao

et al., 2012; Kessler et al., 2020). This process takes place under oxic conditions and produces 2 moles of alkalinity per mole of dissolved $CaCO_3$.

$$CaCO_3 + CO_2 + H_2O \rightarrow Ca^{2+} + 2HCO_3^- \qquad\qquad (6)$$

In anoxic environments, organic matter mineralization generates more $A_T$ than DIC, quickly increasing $\Omega_{cal}$ and thereby inhibiting dissolution (Morse and Mackenzie, 1990; Burdige, 2006). If kiln dusts would be applied to continental shelf waters overlying sediments with potential for enhanced carbonate dissolution, including organic-rich, carbonate-poor marine sediments (Lunstrum and Berelson, 2022; Dale et al., 2024; Biçe et al., 2025; Fuhr et al., 2025) or coastal upwelling zones

(Harris et al., 2013; Fuhr et al., 2025), weathering of all calcite in the residual kiln dust could additionally produce a maximum of 10.8 mmol $A_T$ $g^{-1}$ LKD and 7.4 mmol $A_T$ $g^{-1}$ CKD. However, large-scale fining of sediment with kiln dust could reduce sediment properties, such as the permeability, solute exchange rates, and oxygen penetration depth (Speybroeck et al., 2006; Ahmerkamp et al., 2017). The latter would limit the zone in which metabolic $CaCO_3$ dissolution can occur. Relatively high $CaCO_3$ concentrations may further reduce the dissolution efficiency (i.e. dissolution rate per amount of $CaCO_3$ added) (Dale

et al., 2024). Additionally, ecological impacts may arise through changes of the solid sediment matrix and modifications of porewater conditions (see Sect. 4.3). The potential for enhanced sedimentary alkalinity generation via residual kiln dust addition to organic-rich, carbonate-poor marine sediments therefore warrants further experimental investigation. If fully realized, the total alkalinity release potential (immediate dissolution and metabolic $CaCO_3$ dissolution) could reach up to 18.8 mmol $g^{-1}$ for LKD and 9.8 mmol $g^{-1}$ for CKD. By contrast, in open-ocean applications, the residual material would settle to

the deep seafloor, where metabolic dissolution would occur in waters isolated from the atmosphere and thus would not contribute to CDR on relevant (year–decade) timescales.

## 4.5 Carbon dioxide removal potential

Achieving the Paris Agreement targets will require rapid and deep $CO_2$ emission reductions, complemented by 12–15 Gt $CO_2$ year[-1] of carbon removal by 2100 (Rockström et al., 2017; Minx et al., 2018). Kiln dusts could potentially contribute to this

CDR portfolio. Currently, most kiln dust is landfilled (El-Attar et al., 2017), while the remainder is recycled for applications such as soil stabilization, concrete mix, chemical treatment, ceramics, and brick manufacturing (Al-Bakri et al., 2022). CKD

can replace 5-10% of cement, or up to 20% when combined with pozzolanic materials (fly ash, slag), reducing waste, lowering raw material and energy consumption, and cutting $CO_2$ emissions by a similar percentage (Huntzinger and Eatmon, 2009; Al-Bakri et al., 2022). While this represents the ideal use of CKD in terms of $CO_2$ mitigation, high levels of alkalis, sulfate, and chloride limit the extent to which CKD can be recycled in cement manufacturing (Al-Bakri et al., 2022). Carbonation of kiln dusts, involving the reaction of metal oxides with $CO_2$ to form solid carbonates, has been proposed as an alternative $CO_2$ sequestration method (Huntzinger et al., 2009; Adekunle, 2024):

$$CaO + CO_2 \rightarrow CaCO_3 \tag{7}$$

This process captures 1 mol $CO_2$ per mol metal oxide, which is less than what can be achieved via CaO hydration and subsequent $Ca(OH)_2$ dissociation in seawater (~1.68 mol $CO_2$ $mol^{-1}$ metal oxide). In landfills, both processes naturally occur when kiln dust is exposed to rainwater (Sreekrishnavilasam et al., 2006). However, limited water availability in large kiln dust piles promotes secondary precipitation of carbonates or clay minerals, reducing the effective $CO_2$ sequestration. As such, the ad hoc CDR effect that occurs during landfill disposal of LKD and CKD remain uncertain. Similarly, application of kiln dust to agricultural soils for enhanced weathering purposes (as an alternative to primary mined rocks such as basalt or dunite) could contribute to $CO_2$ removal, though restricted water availability may again increase the risk of secondary mineral formation (Buckingham and Henderson, 2024; Xu and Reinhard, 2025). The focus of this study is the usage of kiln dusts via OAE in natural marine environments. Alternatively, kiln dusts could also be used in reactor-based OAE approaches, such as accelerated weathering of limestone (Rau and Caldeira, 1999; review in Huysmans et al., 2025). These methods allow fast, controlled, and easily monitored alkalinity addition, but require higher energy inputs and dedicated infrastructure compared to ship-based distribution in natural environments (Rau and Caldeira, 1999; Rau et al., 2007; Huysmans et al., 2025).

In this study, short-term weathering of LKD and CKD in seawater produced up to $8.02 \pm 0.53$ and $2.38 \pm 0.16$ mol of alkalinity per kg of source material, respectively (Fig. 2c). On average, 1 mol of added alkalinity sequesters 0.84 mol $CO_2$ in surface ocean waters (Schulz et al., 2023), thus resulting in $297 \pm 20$ g $CO_2$ $kg^{-1}$ LKD and $88 \pm 6$ g $CO_2$ $kg^{-1}$ CKD. With current global annual production of approximately 29 Mt for LKD and 287 Mt for CKD (CEMBUREAU, 2024; USGS, 2025), their maximum carbon dioxide removal potential via dissolution in seawater, assuming an average of 0.07 t of kiln dust produced per tonne of

lime or cement (Al-Refeai and Al-Karni, 1999), is approximately 8.7 ± 0.6 Mt $CO_2$ $yr^{-1}$ for LKD and 25 ± 2 Mt $CO_2$ $yr^{-1}$ for CKD. Additional $CO_2$ uptake through metabolic calcite dissolution in coastal and shelf sediments could potentially further increase this to 13.4 Mt $yr^{-1}$ for LKD and 57 Mt $yr^{-1}$ for CKD, although the effectiveness and time scaling of this process are

uncertain. Cumulatively, this would amount to 1 Gt and 4.3 Gt $CO_2$ by 2100, assuming constant kiln dust production rates and complete utilization for OAE from 2025 onwards. While significant, this represents only 1.9–2.8 % of the 2.5–3.7 Gt $CO_2$ emitted annually by the cement and lime industries (Simoni et al., 2022; Cheng et al., 2023). Therefore, decarbonizing these sectors remains the top priority for effective climate change mitigation (Simoni et al., 2022; Barbhuiya et al., 2024).

The CDR estimates presented here are upper-bound values, assuming that all globally produced LKD and CKD will be used

for OAE, and that production rates remain constant throughout the 21$^{st}$ century. In practice, some kiln dust will be used for other economically viable applications (Al-Bakri et al., 2022), while on the other hand, cement demand is projected to increase by 12–23 % by 2050, which will increase kiln dust production (IEA, 2018). $CO_2$ emissions from transporting kiln dust to the ocean were not considered, but their impact on net CDR is likely minor if deployment occurs near production sites with minimal road transport (Foteinis et al., 2022). These CDR estimates also assume full atmospheric $CO_2$ equilibration of $A_T$-

enriched surface waters ($\gamma_{CO_2} \approx 0.84$), while actual values in coastal regions are possibly lower ($\gamma_{CO_2} \approx 0.65 - 0.8$), due to alkalinity transport to the deep ocean without prior atmospheric exchange (He and Tyka, 2023). Therefore, application should focus on continental shelf seas, especially those with organic-rich, carbonate-poor sediments (Lunstrum and Berelson, 2022; Dale et al., 2024), to promote metabolic $CaCO_3$ dissolution and maximize the CDR potential. Importantly, only one specific type of CKD and LKD were tested in this study, and mineralogical and chemical composition can vary significantly with the

production process, thus affecting the CDR potential (Pavía and Regan, 2010; Siddique, 2014; Drapanauskaite et al., 2021). Tailoring application concentrations to site-specific conditions and material properties is therefore essential for safe and effective deployment.

## 5    Conclusions

Cement kiln dust (CKD) and lime kiln dust (LKD) are abundant, fine-grained, and alkaline industrial byproducts available at

low cost. We conducted laboratory experiments to evaluate their suitability as mineral-based OAE feedstocks. Reactive phases

in CKD and LKD dissolved rapidly in continuously stirred seawater, with 65–92 % dissolving within 1 hour and complete reactive phase dissolution within 24 hours. LKD generated up to 8.02 ± 0.53 mmol of alkalinity per g, compared to 2.38 ± 0.16 mmol g⁻¹ for CKD. Alkalinity consuming secondary aragonite precipitation was observed at saturation states ≥6.2, with a saturation state of ~5 identified as a safe threshold. Based on current global production (~29 Mt year⁻¹ for LKD and ~287 Mt year⁻¹ for CKD), the theoretical maximum CDR potential via dissolution in seawater is 8.7 ± 0.6 Mt $CO_2$ year⁻¹ for LKD and 25 ± 2 Mt $CO_2$ year⁻¹ for CKD. Turbidity increases from both LKD and CKD, and trace metal release from CKD, present potential environmental risks. To minimize secondary mineral formation and ecological impacts, site-specific application concentrations should be determined through particle dispersal modelling that accounts for local hydrodynamic conditions. A large portion of both materials, 75 % of LKD and 71 % of CKD, remained undissolved, with calcite making up 72 % and 52 % of this residual fraction, respectively. If this residual calcite undergoes metabolic dissolution in marine sediments, it could further contribute to CDR, potentially adding up to 4.7 Mt $CO_2$ year⁻¹ for LKD and 32 Mt $CO_2$ year⁻¹ for CKD, although this requires further experimental validation. Overall, LKD and, to a lesser extent, CKD show promise for OAE, with a CDR potential of up to 13.4 Mt year⁻¹ for LKD and 57 Mt year⁻¹ for CKD at current production levels, based on our small-scale laboratory experiments. However, additional experiments that more closely mimic natural conditions are warranted to further constrain particle behaviour in the water column, interactions with sediments, and potential biological impacts of kiln-dust–based OAE.

**Appendix A: Kiln dust properties**

**A1 Grain size, specific surface area, and elemental composition**

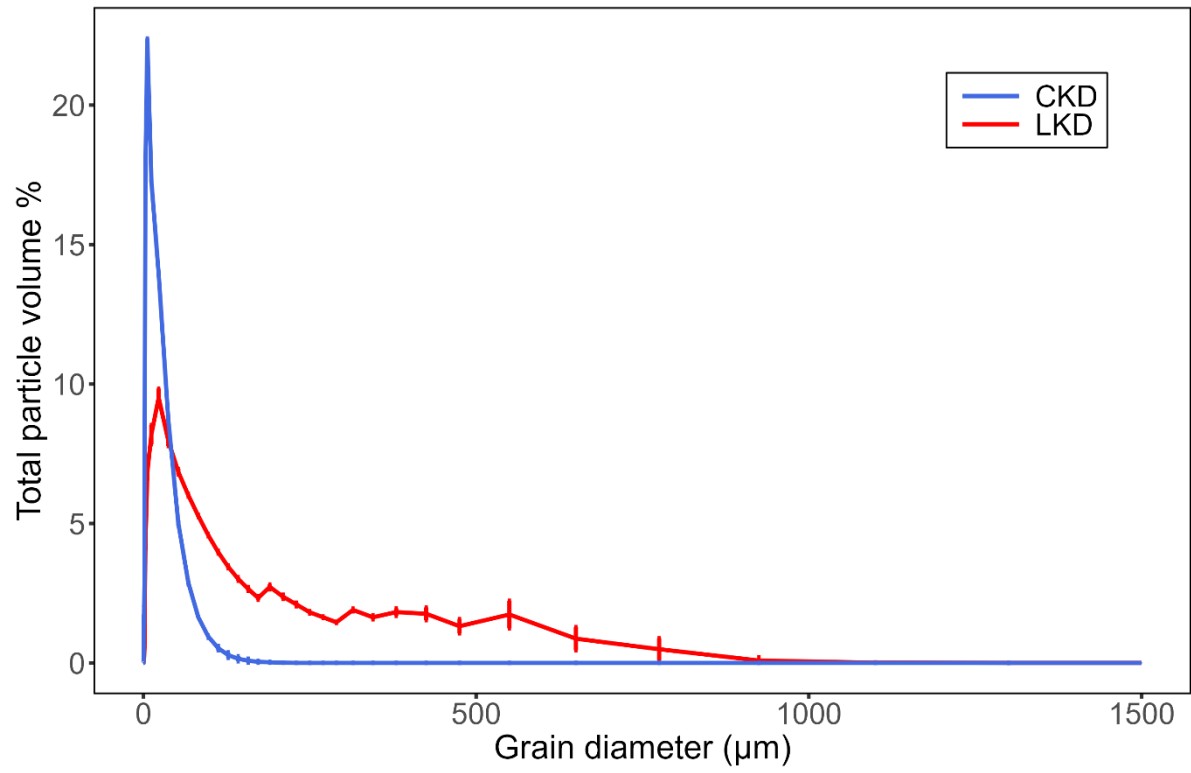


**Figure A1: Volumetric grain size distribution of the fresh lime kiln dust (LKD, in red) and cement kiln dust (CKD, in blue). Mean and standard deviation are shown (N = 3).**

The geometric surface area $A_{GEO}$ (m² g⁻¹) of the experimental kiln dusts is shown in Table 2 of the main text. It was calculated from the different grain diameter classes i (n = 37, between 0.35 and 1300 µm grain diameter) of the volumetric particle size distribution (Fig. A1) as

$$A_{\text{GEO}} = \sum_{i=1}^{n} \left( \frac{\frac{V_{KD_i}}{V_{grain_i}} * A_{grain_i}}{\rho_{KD}} \right) \tag{1}$$

Where $V_{KD_i}$ represents the relative volume (cm³ cm⁻³) for a certain grain diameter class $i$ (e.g. 180 – 200 µm). $V_{grain_i}$ and

$A_{grain_i}$ are the volume (cm³) and surface area (m²) of a single KD grain calculated from the average grain diameter of a certain grain diameter class (e.g. 190 µm for 180 – 200 µm), assuming perfect spherical particles. $\rho_{KD}$ is the specific gravity of the kiln dust (g cm⁻³).

**Table A1: Elemental composition (wt%) of the cement kiln dust (CKD) and lime kiln dust (LKD). Concentrations were analysed via**
**ICP-OES after heated digestion in a mixture of HClO₄, HNO₃, and HF as described in section 2.1 of the main text.**

| Element composition (wt%) | CKD | LKD |
| --- | --- | --- |
| Al | 3.02 | 0.038 |
| As | <0.000016 | <0.000016 |
| Ba | 0.044 | 0.00082 |
| Be | 0.000095 | 0.000012 |
| Ca | 27.8 | 44.9 |
| Cd | 0.0030 | <0.00000085 |
| Ce | 0.0025 | <0.0000028 |
| Co | 0.0015 | <0.0000021 |
| Cr | 0.017 | 0.0011 |
| Cu | 0.028 | 0.0015 |
| Fe | 2.44 | 0.11 |
| K | 6.05 | 0.17 |
| Li | 0.0079 | 0.00025 |
| Mg | 0.57 | 0.22 |
| Mn | 0.11 | 0.0049 |
| Mo | <0.0000082 | <0.0000082 |
| Na | 1.22 | 0.080 |
| Ni | 0.011 | <0.0000043 |
| P | 0.12 | <0.000011 |
| Pb | 0.15 | <0.000014 |
| S | 4.64 | 0.63 |
| Sc | 0.00046 | <0.00000008 |
| Sr | 0.072 | 0.028 |
| Ti | 0.19 | 0.0013 |
| V | 0.036 | 0.00063 |
| Y | 0.0013 | 0.00016 |
| Zn | 0.65 | 0.0029 |

**A2 Kiln dust settling time**

To provide an initial simplified assessment of whether kiln dust particles could settle out of the ocean's mixed layer (assumed to be 200 m deep) before the complete dissolution of their reactive phases, Stokes' law (Eq. (2)) was applied to estimate their gravimetric settling velocity, assuming spherical particle geometry:

$$v = \frac{gd^2(\rho_p - \rho_s)}{18\eta} \qquad (2)$$

where $g$ is the acceleration of gravity (9.81 m² s$^{-1}$), $d$ is the particle diameter (m), $\rho_p$ is the density of the particle (2872 kg m$^{-3}$ for LKD and 2712 kg m$^{-3}$ for CKD), $\rho_s$ is the density of the solution (1022 kg m$^{-3}$ for 32 ‰ seawater at 20°C), and $\eta$ is the dynamic viscosity of the solution (0.00108 kg m$^{-1}$ s$^{-1}$). Seawater density and dynamic viscosity were derived using the "swRho" and "swViscosity" function of the "oce" package in R. The settling time (h) required for particles to exit the mixed layer was

calculated by dividing the mixed layer depth (200 m) by the settling velocity and converting the result from seconds to hours by multiplying by 3600.

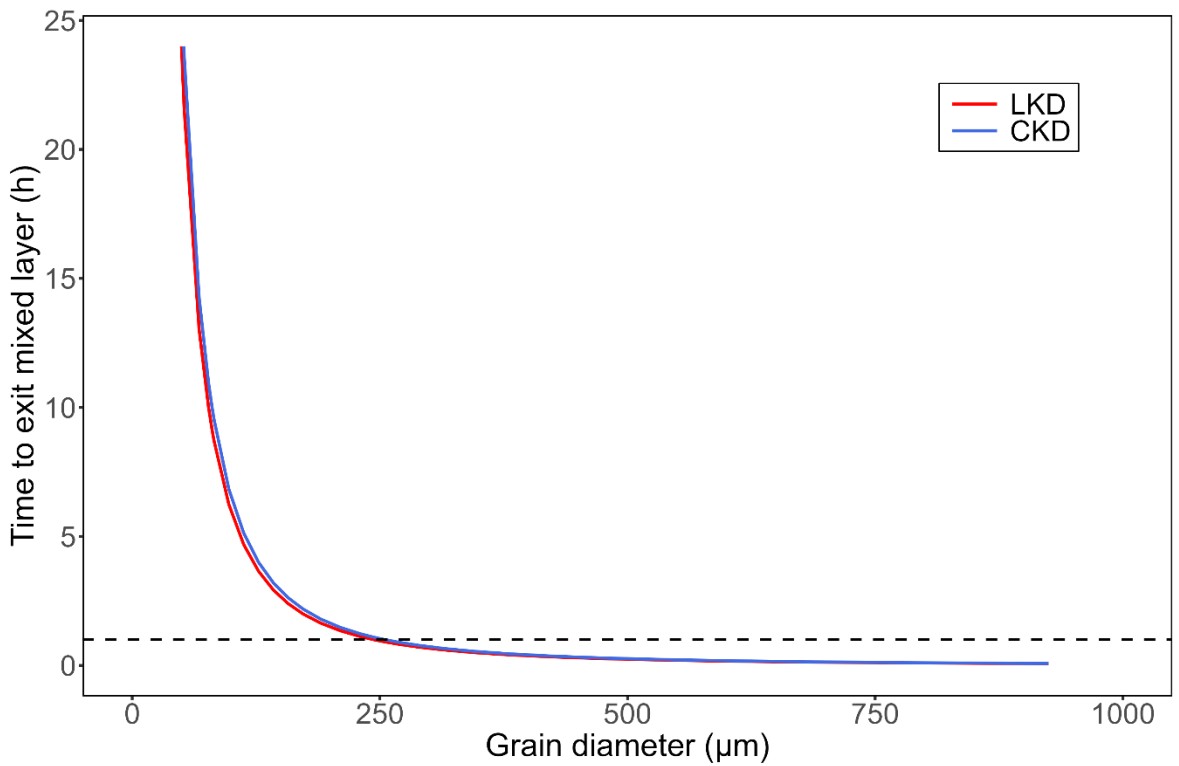

**Figure A2: Expected time for perfectly spherical lime kiln dust (LKD, in red) and cement kiln dust (CKD, in blue) particles to sink below the mixed layer depth (assumed to be 200 m) according to Stokes' law. Horizontal black dashed line at represents the time (1 h) at which most (72 – 85%) reactive phases in LKD and CKD have dissolved.**

After one hour of incubation at low application concentrations, most of the reactive phases in CKD (72 ± 12%) and LKD (85 ± 11%) had dissolved (Fig. 1b). Particles of LKD smaller than 244 µm and CKD smaller than 255 µm will remain suspended in the mixed layer for at least one hour (Fig. A2). Given the measured grain size distributions (Fig. A1), this means that 100% of CKD particles and 85 ± 2% of LKD particles would stay suspended in a 200 m mixed layer for this duration.

**A3 Residual kiln dust fraction experiment**

Due to the limited remaining material (<70 mg) after the dissolution experiments presented in the main text, it was not possible to accurately quantify the residual mass of kiln dusts following dissolution in seawater. To address this, a separate small-scale test was conducted to determine the residual mass fraction. Two 2-liter plastic bottles were filled with 2 L of filtered seawater and continuously aerated using air stones to speed up equilibration with atmospheric $CO_2$. Kiln dust was added to each bottle at concentrations of 87 mg kg⁻¹ for CKD and 21 mg kg⁻¹ for LKD, three times per day over the course of three consecutive days. Additions were spaced a minimum of 2.5 hours apart to avoid high $pH_T$ increases that might induce secondary aragonite formation. After the final addition, the bottles were left to incubate at room temperature (16.1–17.5 °C) for 24 hours to ensure

complete dissolution of the reactive phases. The suspensions were then filtered through pre-weighed dried (at 60 °C for 24 h)

0.3 μm pore size membrane filters (Seitz type M) using a Sartorius Microsart E-jet filtration unit. The filters were placed in pre-weighed Al foil cups, then dried at 60 °C for 72 hours, and subsequently reweighed using the same Mettler Toledo XP26 Excellence Plus microbalance to determine the residual solid mass. For LKD, the residual fraction was 75.42% of the added mass (379.28 mg), while for CKD, it was 70.58% of the added mass (1567.65 mg).

## Appendix B: Seawater parameters experiment II

**B1 Observed seawater chemistry changes**

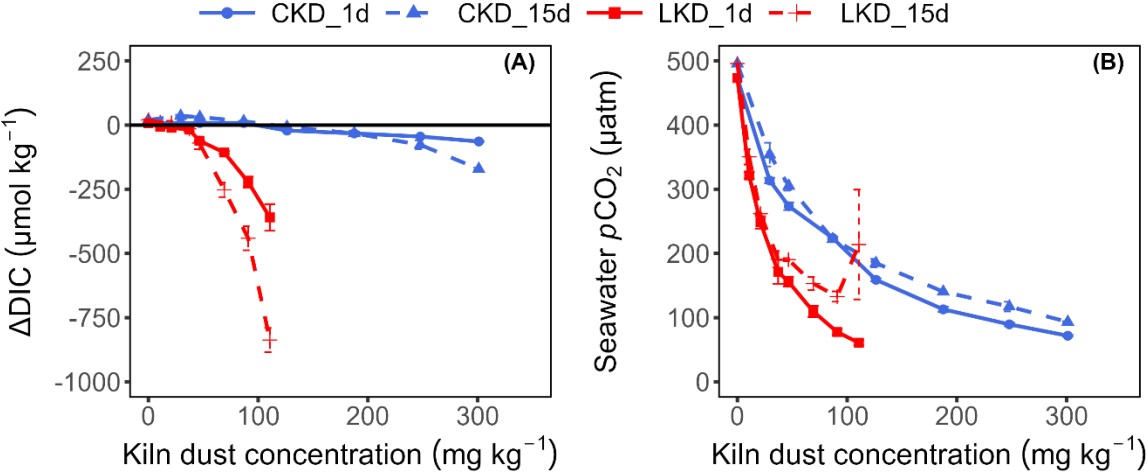

**Figure B1. (A) Change in DIC concentration (μmol kg⁻¹) and (B) partial CO₂ pressure (*p*CO₂, expressed in μatm) as a function of the kiln dust application concentration (mg kg⁻¹) during dissolution experiment II. Results (as mean ± S.D., N=3) are shown for lime kiln dust (LKD, in red) and cement kiln dust (CKD, in blue) after 1 day (solid lines) or 15 days of incubation (dashed lines) in FSW.**


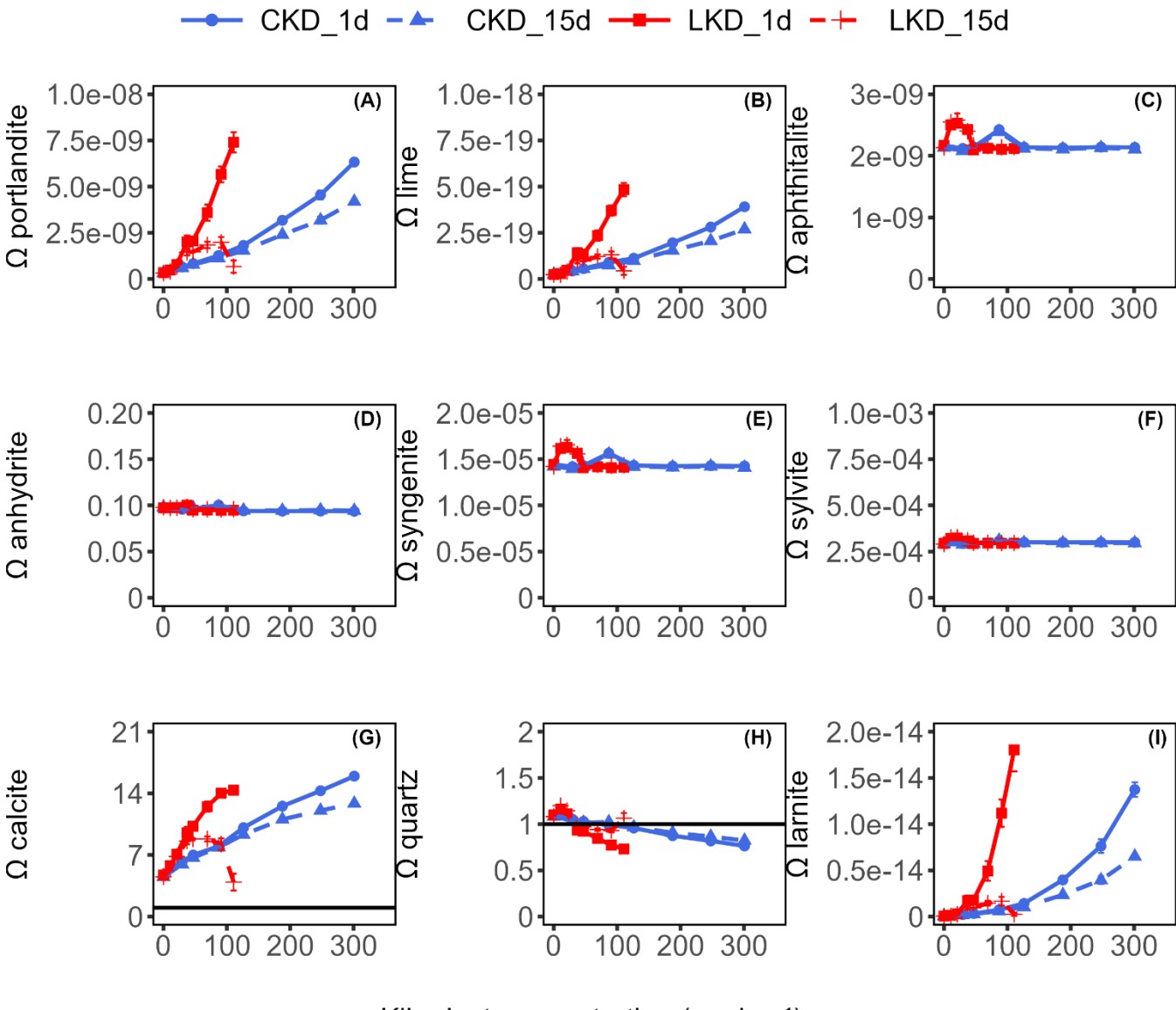

**Figure B2.** Saturation state (Ω) of (A) portlandite, (B) lime, (C) aphthitalite, (D) anhydrite, (E) syngenite, (F) sylvite, (G) calcite, (H) quartz, and (I) larnite as a function of the kiln dust application concentration (mg kg$^{-1}$) during experiment II. Results (as mean ± S.D., N=3) are shown for lime kiln dust (LKD, in red) and cement kiln dust (CKD, in blue) after 1 day (solid lines) or 15 days of incubation (dashed lines) in FSW. The horizontal solid black line at Ω = 1 represents the critical saturation threshold.


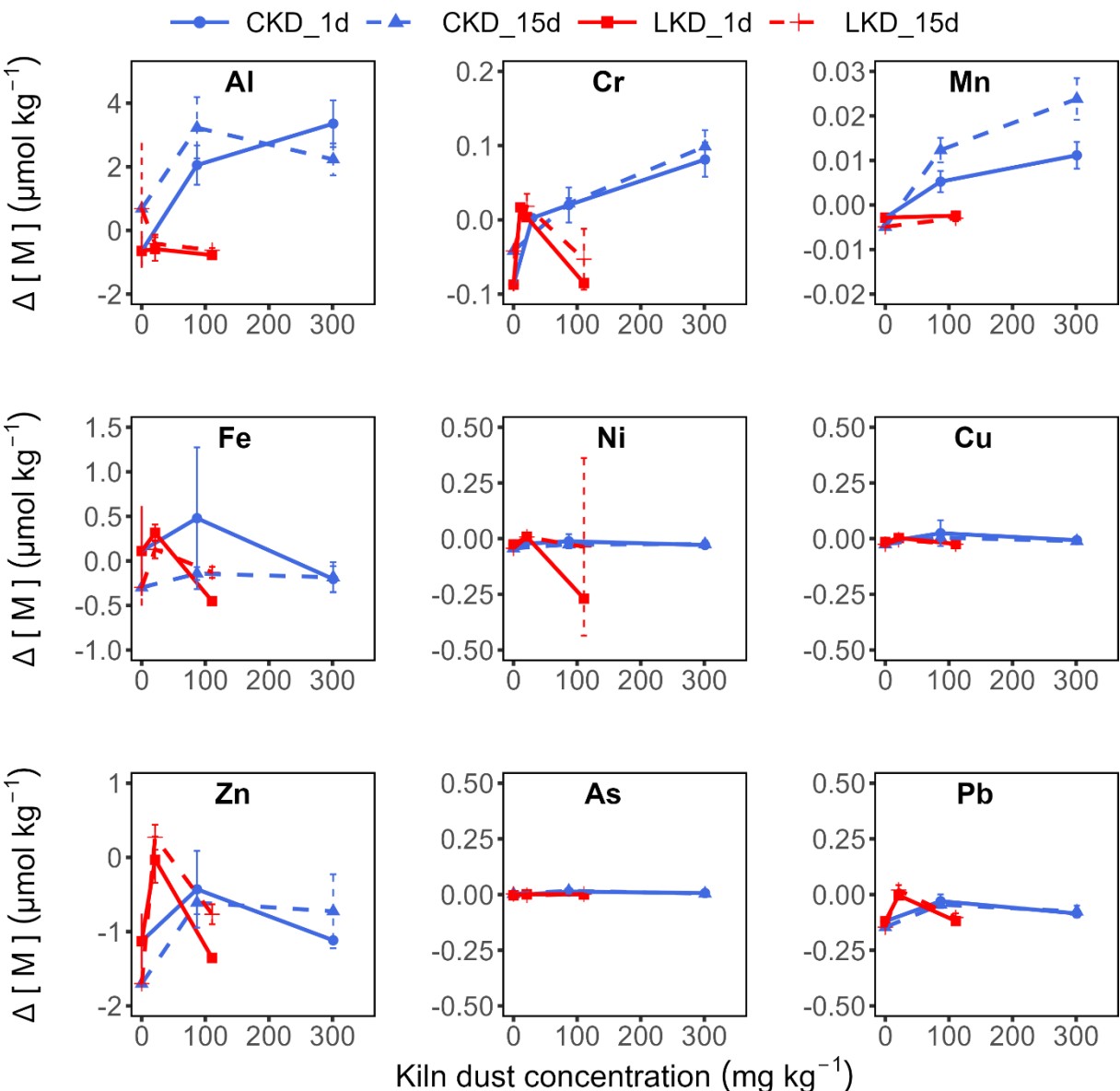

Figure B3: Metal(loid) accumulation in filtered seawater (µmol kg$^{-1}$) as a function of the kiln dust application concentration (mg kg$^{-1}$) during dissolution experiment II. The accumulation was calculated as the increase in metal(loid) concentrations above the initial seawater levels. Results (as mean ± S.D., N=3) are shown for lime kiln dust (LKD, in red) and cement kiln dust (CKD, in blue) after 1 day (solid lines) or 15 days of incubation (dashed lines) in FSW.

**B2 Expected seawater trace metal accumulation**

The expected accumulation $\Delta C_i$ (expressed in µmol kg$^{-1}$) of a given metal(loid) $i$ in filtered seawater (FSW) under the assumption of stoichiometric KD weathering was calculated from the measured total metal(loid) concentration $\chi$ (expressed in mg kg$^{-1}$) in the KD and the dissolved KD fraction $\varphi_{diss}$ (dimensionless) as follows:

$$\Delta C_i = \frac{m_{KD}\varphi_{diss}\chi}{m_{FSW}M_i} \tag{3}$$

Where $m_{KD}$ and $m_{FSW}$ reflect the masses (g) of kiln dust and filtered seawater, respectively, used in the plastic incubation vials. $M_i$ denotes the molar mass of a given metal(loid) $i$.

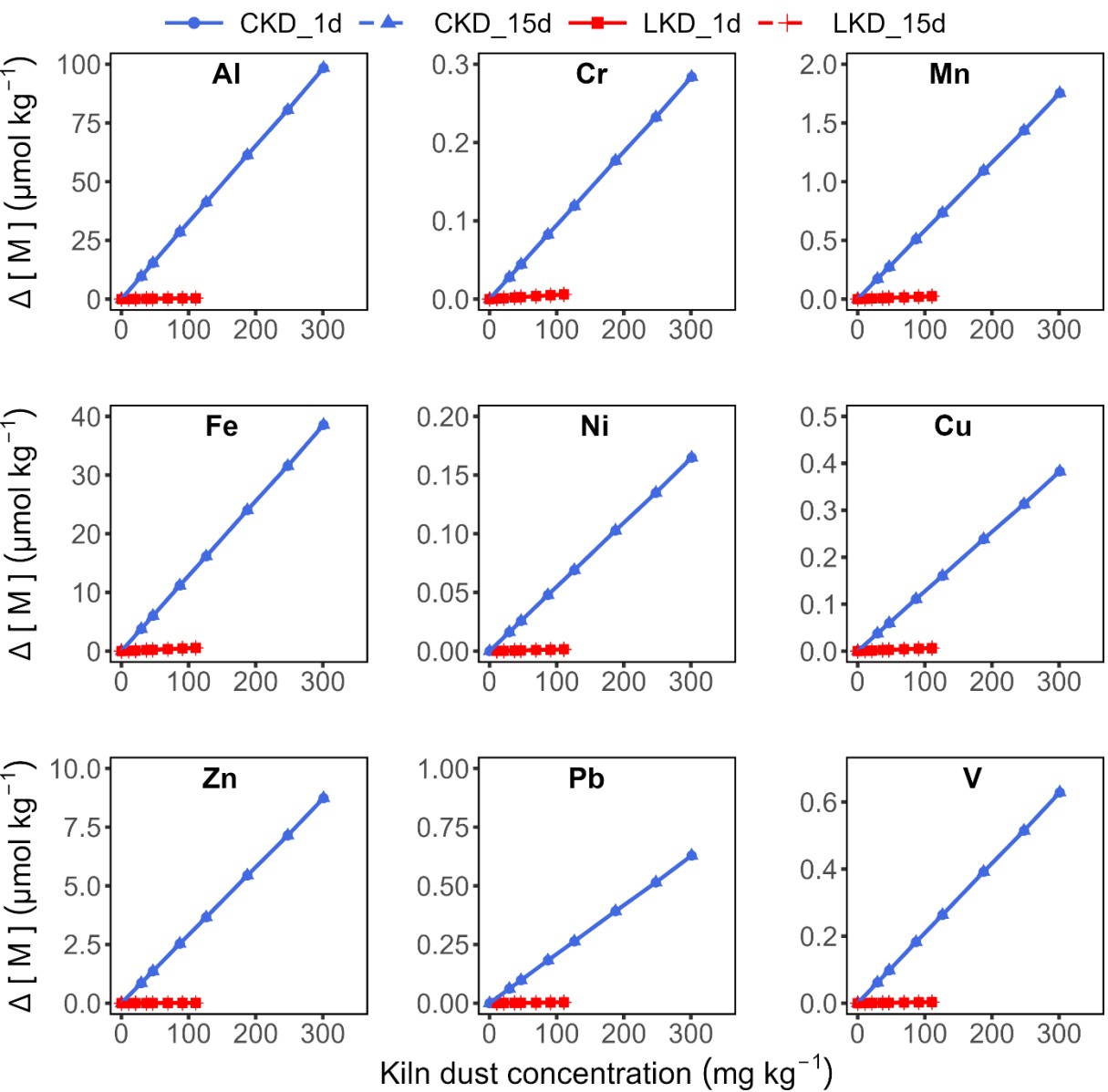

**Figure B4. Expected metal(loid) accumulation in filtered seawater (μmol kg$^{-1}$) as a function of the kiln dust application concentration (mg kg$^{-1}$) during dissolution experiment II. Results are shown for lime kiln dust (LKD, in red) and cement kiln dust (CKD, in blue). Accumulation values after 1 day (solid lines) and 15 days (dashed lines) were assumed to be equal given that all total alkalinity producing reactive phase dissolution occurred within the first day.**

## Appendix C: Maximum sedimentary kiln dust application concentration

In real-world ship-based deployments, residual kiln dust (KD) would settle on the seafloor, where it would be mixed into the surface sediment through local hydrodynamics and bioturbation. This accumulation and possible subsequent dissolution of residual KD, could lead to the build-up of metals in the surface sediment, potentially posing a toxicological risk to benthic organisms. Sediment Quality Guidelines (SQGs) are employed to assess the risk of metal toxicity to marine biota in a tiered approach, with the first step involving the comparison of total sediment metal concentrations to these guidelines (Hübner et al., 2009; Simpson and Batley, 2016). The kiln dust contain a range of metals (Appendix A Table A1) of which Pb could mostly easily exceed existing SGQs based on preliminary screening. Following Flipkens et al. (2021) we derived the maximum allowable KD application $m_{app}$ (expressed in kg m$^{-2}$ seafloor) that would not exceed marine Pb SQGs via:

$$m_{app} = \frac{(C_{SQG} - C_{bg})V_s \rho_s (1 - \Phi)}{\chi} \qquad (4)$$

where $C_{SQG}$ is the sediment quality guideline for Pb (mg kg$^{-1}$ dry wt), $C_{bg}$ is the background sedimentary Pb concentration (mg kg$^{-1}$ dry wt), $V_s$ is the volume of the sediment in which the kiln dust is mixed per m$^2$ of seabed (m$^3$ m$^{-2}$ seabed), $\rho_s$ is the specific density of marine sediment (2650 kg m$^{-3}$), $\Phi$ is the porosity of the sediment, and $\chi$ is the concentration of Pb (mg kg$^{-1}$ dry wt) in CKD or LKD (Table A1). The porosity of marine surface sediment was assumed to be 0.60 based on the predicted global coastal sediment porosity ranging from approximately 0.50 to 0.85 (Martin et al., 2015). A sediment mixing depth of 10 cm was assumed, reflecting the typical depth where most benthic biota are found (Simpson and Batley, 2016; Solan et al., 2019). Given the global variation in Pb SQGs, maximum allowable KD application ($m_{app}$) calculations were made using different sediment quality guidelines, including the Chinese Interim Sediment Quality Guideline–Low (ISQV-low), the Australian Interim Sediment Quality Guideline–Low (ISQG-low), the Norwegian Predicted No Effect Concentration (PNEC), and the American Threshold Effect Level (TEL) (Hübner et al., 2009).

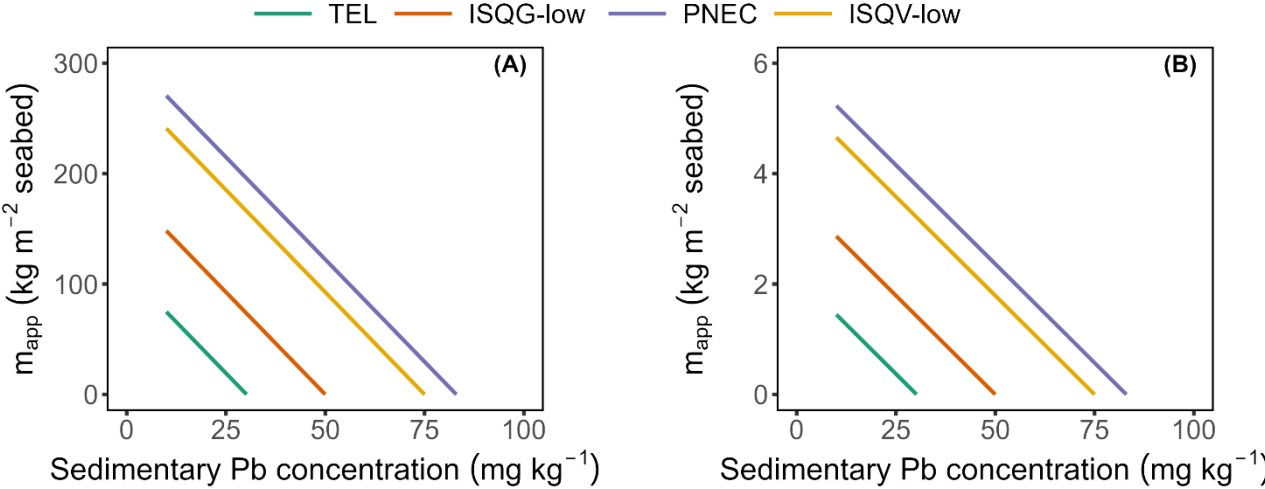

**Figure C1: Maximum allowable application ($m_{app}$, expressed in kg m$^{-2}$) of (A) lime kiln dust or (B) cement kiln, without exceedance of lead (Pb) sediment quality guidelines (SQGs), plotted as a function of the background sedimentary Pb concentration (mg kg$^{-1}$). The considered SQGs include the Threshold Effect Level (TEL, green), the Interim Sediment Quality Guideline–low (ISQG-low, orange), the Predicted No Effect Concentration (PNEC, blue), and the Interim Sediment Quality Value–low (ISQV-low, yellow). A sediment mixing depth of 10 cm was assumed, and kiln dust Pb concentrations were based on measured values reported in Table A1. Results represent a conservative, worst-case scenario in which all Pb released during KD dissolution is retained within the sediment.**

## Code availability

The code used for analysis in this study is available upon request. Interested parties may contact the corresponding author.

## Data availability

The data supporting the findings of this study have been deposited in Zenodo, DOI: https://doi.org/10.5281/zenodo.17938383.

## Author contribution

GF was responsible for conceptualization, data curation, formal analysis, investigation, methodology, visualization, writing original draft preparation, and writing review and editing. GF and GL carried out the investigation. FJRM was responsible for conceptualization, methodology, funding acquisition, resources, supervision, and writing review and editing.

**Competing interests**

The authors declare that they have no conflict of interest.

**Acknowledgements**

The authors thank Afshin Neshad Ashkzari and Helen de Waard (Utrecht University) for conducting the ICP-OES analyses, Max Van Brusselen (UAntwerpen) for the ICP-MS analysis, and Tom Van Gerven and Michèle Vanroelen (KU Leuven) for the BET analysis. Moreover, we appreciated the help of Romello Cavalier (UAntwerpen) with the sample collection and we thank Tom Huysmans (UAntwerpen) for diluting the seawater samples for ICP-MS analysis. Finally, during the preparation of this work, the authors used ChatGPT to assist in improving the flow of the text. All content was subsequently reviewed and edited by the authors, who take full responsibility for the final content of the publication.

**Financial support**

This research was supported by the VLAIO De Blauwe Cluster project "Blue Alkalinity" (HBC.2023.0496).

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
