# Peer review of "The carbon dioxide removal potential of cement and lime kiln dust via ocean alkalinity enhancement"

_EGUsphere, 2025_

## Referee Comment (RC2)

Reviewer comments – egusphere-2025-4887 – The carbon dioxide removal potential of cement and lime kiln dust via ocean alkalinity enhancement

**Overview:**

The manuscript from Flipkens et al. explores the suitability of two by-products of the cement industry, cement and lime kiln dusts (CKD and LKD, respectively). Using laboratory experiments, they report first estimates on the dissolution kinetics of such feedstocks in natural seawater, both using short- and long-term experiments, while expanding on the risks for CaCO3 precipitation and potential trace metals release. Finally, CDR estimates are given, assuming the use of all available CKD and LKD, and considering inorganic thresholds and water quality and safety guidelines.

Overall, I found the manuscript very enjoyable to read. While some people may express reluctance to consider such by products for OAE, their significant alkalinity release makes them suitable candidates, and the presented data support this. The introduction is rather comprehensive, and most required information is already provided. The material and method section is also mostly complete with some smaller comments pointed out below. The results section is rather dense, yet all required information is well presented. Finally, the discussion is convincing and reports well on the implication of kiln dust based OAE. Some further points could be added to the discussion, especially when it comes to the alkalinity generation. While LKD was following estimates, the nearly doubling alkalinity generation from CKD is yet to be fully addressed. I believe a more detailed discussion could benefit the paper. It would be beneficial to try and characterise the amorphous phases that clearly seem to be the responsible factor. I do not request further analyses but rather consider whether using the elemental composition of CKD (Table A1), an estimate of the amorphous phases' composition could be derived. Another point that could benefit from further interpretation is the fate of KD and especially the potential secondary minerals precipitated. It is clear that non soluble phases such as CaCO3 would eventually sink and as described, potentially dissolve on the seabed. It would also be the case for secondary CaCO3 formed from the high alkalinity generated. One could consider discussing the dissolution on the seabed of both the CaCO3 from KD and the freshly precipitated CaCO3 in the water column. The overall potential would probably increase, and if it is significant, it would be worth mentioning. Finally, a smaller point regarding the turbidity discussion. Discussing such environmental guideline is a great idea and of high importance. Unless I am mistaken, the turbidity is reported for 24h. In line 343, it is mentioned that the KD particles would remain in the surface ocean mixed layer for about one hour. Therefore, one could safely assume that after 1h the turbidity is back to low enough levels that new KD could be added. Under such assumption, the overall KD release could be increased, ultimately increasing the CDR potential. If it is correct, it could be quicky discussed.

Considering the minor comments mentioned above, and once they have been addressed, I am fully supporting the publication of the manuscript.

**Comments:**

Line 6: I am not sure whether "ocean liming" is a suitable keyword here, considering it investigates kiln dust potential for OAE rather than lime. But this can be up to the authors.

Line 34: I believe brackets are missing and should read "brucite (Mg(OH)2)".

Line 42: one could argue the use of "rapid" here, as it can take month to years before the alkalised water equilibrates with atmospheric CO2. Please edit accordingly.

Line 57: consider moving "hence" at the start of the sentence.

Line 69: has this method been tested and proven to be reliable and efficient at dissolving such product? If so, a reference (or in-house quality control) could be inserted.

Lines 87-88: how long was the seawater aerated for? Was there any check to confirm equilibration? (pH measurements, pCO2, etc.).

Line 111: how were DIC samples taken? Given the gas sensitivity of these samples, it would be interesting to report whether a peristaltic pump was used or any other apparel.

Line 125-126: since the pH has been measured and later reported on the total scale, it would be beneficial to report it as  $pH_T$  throughout the text

Line 150: salinity has no unit, please remove the "ppt". alternatively, you could report the ionic strength instead, but it is not required

Line 164: I believe PHREEQC provides saturation indexes, not saturation states. The term  $\Omega$  is usually used only for CaCO3 if I am correct

Lines 164-169: I am not sure I understand correctly. First it is said that saturation states were calculated with PHREEQC (line 164). Then, it is said that  $\Omega_A$  and  $\Omega_C$  were not computed in PHREEQC. Why that?

Line 246: I believe that "for CKD is missing after the 1.3 mmol g-1.

Line 264: it would be great to have the 0 on the y axis of tile B.

Lines 293-294: how can turbidity be slightly greater yet significantly greater? Do you mean statistically significantly greater?

Line 310: it would be great to have the 0 on the y axis of tile B.

Line 330: I believe "be" should be deleted.

---

## Author Comment (AC2)

**Referee #1**

The authors present experimental results on the reactivity of kiln dust in seawater. Kiln dust is a waste product of the cement and lime industry that is either recycled or disposed in e.g. landfills. The authors show that reactive phases in kiln dust (CaO, Ca(OH)2) are rapidly dissolved in seawater such that most of these phases are consumed within a few minutes after exposure to seawater. Kiln dust added to marine surface waters would, hence, add alkalinity and promote CO2 uptake in the surface ocean. The remaining material (mostly calcite) may settle to the seabed where it might produce additional alkalinity if sediment porewaters are undersaturated with respect to calcite.

The experimental results are very solid and clearly show the potential of kiln dust for ocean alkalinity enhancement (OAE). The paper is well written and I would suggest to publish the paper after a few revisions that are outlined below:

We sincerely thank the reviewer for the thorough and thoughtful evaluation of our manuscript. We appreciate the time and effort dedicated to carefully reading our work and providing constructive feedback which helped us improve the quality and clarity of the paper.

The authors present the mineral composition of kiln dusts in Table 2. It would be good to add data on Ca-Si-phases that are formed during cement production and should be present in cement kiln dust (CKD). These cement phases should have a high reactivity in seawater. They could explain the release of excess alkalinity observed in CKD experiments (line 331). The authors should consider these Ca-Si-phases in the discussion of their experimental results and add information on these phases in Table 2 if possible.

Calcium silicates are indeed relevant when discussing the alkalinization potential of CKD. However, as noted in the review by Adekunle (2024), these phases are typically present in smaller quantities compared to lime and portlandite. In our study, we conducted quantitative XRD analyses in duplicate. No Ca–Si phases were detected in the first measurement, while the second measurement indicated the possible presence of small amounts of beta-dicalcium silicate ( $\beta$ -C2S 2.1%), and also of the iron oxides hematite (0.68%), and maghemite (0.56%). The identification of these phases was uncertain, which is why they were not included in Table 2 summarizing the mineralogical composition. Still, we follow the valuable suggestion of the referee, and we now additionally discuss how calcium silicates can contribute to alkalinity generation upon CKD dissolution in seawater. So, we now discuss their potential role in the text (shown in blue below, with changes highlighted in yellow) and we also indicate in the header of Table 2 that these phases were detected with low confidence.

Line 184: Furthermore, minor phases including 2.1% larnite/ $\beta$ -C2S (Ca2SiO4), 0.68% hematite (Fe2O3), and 0.56% maghemite ( $\gamma$ -Fe2O3) were identified with low confidence in one of the duplicate CKD samples analyzed by XRD.

Line 197: Furthermore, the hematite and maghemite present in the CKD are essentially insoluble under oxic conditions in natural seawater, and filtration would remove Fe-reducing bacteria capable of enhancing their dissolution (Canfield, 1989). In contrast, the dissolution of portlandite and lime each produces two moles of AT per mole (Eq. 3), while larnite (potentially present in the CKD) would yield four moles of AT per mole upon dissolution (Brand et al., 2019). Overall, complete dissolution of these phases corresponds to maximum alkalinity contributions of 8.8 mmol g-1 for the LKD and 1.7 mmol g-1 for the CKD, respectively.

Line 198: Physicochemical properties of the experimental cement kiln dust (CKD) and lime kiln dust (LKD). ND indicates that the phases were not detectable. The minor phases larnite/beta-calcium

disilicate ( $\beta$ -C2S; 2.1%), hematite (Fe2O3; 0.68%), and maghemite ( $\gamma$ -Fe2O3; 0.56%) were detected with low confidence in one of the duplicate samples and are therefore mentioned here but not included in the table. The complete measured elemental composition is provided in Appendix A Table A1.

Table 2: theoretical alkalinization potential of CKD was changed to 1.7 mmol  $g^{-1}$  to account for  $A_T$  production by larnite dissolution.

Line 324: CKD is more compositionally complex, typically containing calcite along with various sulfates, chlorides, silicates, and aluminates, including belite, aphthitalite, spurite, ettringite, arcanite, and ferrite (Ayman et al., 2004; Siddique and Rajor, 2012; Beltagui et al., 2017; Adekunle, 2024; Lee and Choi, 2024; Nikolov et al., 2025).

Line 338: Calcium silicates (e.g. larnite  $Ca_2SiO_4$ ) are also alkalinity-generating phases that occur in minor amounts in CKD. They originate form the raw materials used in cement production (e.g. iron ore, clay, or shale) and exhibit a relatively high reactivity in water (Brand et al., 2019; Adekunle, 2024). Dissolution of larnite present in our CKD sample ( $^{\sim}$  2.1 %) could therefore further account for 17 ± 1% of the observed  $A_T$  release. The remaining  $^{\sim}$ 29% of the alkalinity released from CKD likely originated from dissolution of amorphous phases, including (partially dehydrated) clay minerals, reactive amorphous silica, and kiln-derived materials such as fly ash or slag (Khanna, 2010; Pavía and Regan, 2010)

Line 572: Figure B2. Saturation state ( $\Omega$ ) of (A) portlandite, (B) lime, (C) aphthitalite, (D) anhydrite, (E) syngenite, (F) sylvite, (G) calcite, and (H) quartz, and (I) larnite as a function of the kiln dust application concentration (mg kg-1) during experiment II.

The authors propose that kiln dust should be applied to the surface ocean in shelf regions where the seabed is covered by permeable (sandy) sediments to allow for calcite dissolution in sediments that would add further alkalinity to the ocean (line 420). It is, however, likely that porewaters of these permeable sediments have a composition that is close seawater due to the rapid exchange with ambient bottom waters driven by fast tidal currents. Since shelf waters are usually oversaturated with respect to calcite, calcite dissolution may not proceed in these permeable deposits. Muddy sediments with restricted advective porewater exchange might offer a better environment for respiration-driven calcite dissolution as discussed in Dale et al., 2024 and Fuhr et al., 2025. I would suggest to update the text accordingly considering that muddy deposits are at least as favorable for calcite dissolution as permeable (sandy) sediments.

We agree that the statement regarding preferred sediment types for kiln dust application should be more carefully formulated. The suitability of a given sediment strongly depends on the local degree of calcium carbonate undersaturation in the porewater, which in turn is controlled by multiple factors, including the rates of aerobic mineralization, oxidation of reduced compounds, and pre-existing CaCO3 dissolution, as well as temperature and salinity. Because these parameters can vary substantially across spatial scales, local sediment characteristics should ideally be assessed when selecting kiln dust application sites.

Nevertheless, it is evident that organic-rich, carbonate-poor sediments represent the most favorable environments for enhanced carbonate dissolution. Furthermore, coastal upwelling zones have been proposed by Fuhr et al. (2025) as promising application sites due to their high-pCO2 seawater and high mineralization rates. This clarification has been incorporated into the revised text as follows:

Line 429: If kiln dusts would be applied to continental shelf waters overlying sediments with potential for enhanced carbonate dissolution, including organic-rich, carbonate-poor marine sediments

(Lunstrum and Berelson, 2022; Bice et al., 2024; Dale et al., 2024; Fuhr et al., 2025) or coastal upwelling zones (Harris et al., 2013; Fuhr et al., 2025), weathering of all calcite in the residual kiln dust could additionally produce a maximum of 10.8 mmol  $A_T$   $g^{-1}$  LKD and 7.4 mmol  $A_T$   $g^{-1}$  CKD.

Line 435: The potential for enhanced sedimentary alkalinity generation via residual kiln dust addition to organic-rich, carbonate-poor marine sediments therefore warrants further experimental investigation

Line 484: Therefore, application should focus on continental shelf seas, especially those with organic-rich, carbonate-poor sediments (Lunstrum and Berelson, 2022; Dale et al., 2024), to promote metabolic CaCO₃ dissolution and maximize the CDR potential.

Kiln dust disposed in landfills reacts with CO2-bearing rain waters which may lead to a substantial uptake of atmospheric CO2. It is not clear to me whether the total CO2 uptake is enhanced when this material is added to the ocean instead of being disposed on land. The authors should add a paragraph on cement and kiln dust weathering under terrestrial conditions which has been intensively studied over the past decades. They should also try to compare the net CO2 balance of their approach (using kiln dust for OAE) with alternative kiln dust uses (disposal on land, recycling).

We fully agree with this suggestion. The CO2 uptake from using kiln dust for ocean alkalinity enhancement is now briefly compared to terrestrial applications, such as landfill disposal, enhanced weathering, and recycling, in the discussion section on carbon dioxide removal potential. Additionally, the potential benefits and drawbacks of using kiln dust use in a reactor type environment, such as in accelerated weathering of limestone or ship-based ocean deployment, are now addressed to respond to the community comment by Ken Caldeira.

Line 442: Achieving the Paris Agreement targets will require rapid and deep CO2 emission reductions, complemented by 12–15 Gt CO2 year-1 of carbon removal by 2100 (Rockström et al., 2017; Minx et al., 2018). Kiln dusts could potentially contribute to this CDR portfolio. Currently, most kiln dust is landfilled (El-Attar et al., 2017), while the remainder is recycled for applications such as soil stabilization, concrete mix, chemical treatment, ceramics, and brick manufacturing (Al-Bakri et al., 2022). CKD can replace 5-10% of cement, or up to 20% when combined with pozzolanic materials (fly ash, slag), reducing waste, lowering raw material and energy consumption, and cutting CO2 emissions by a similar percentage (Huntzinger and Eatmon, 2009; Al-Bakri et al., 2022). While this represents the ideal use of CKD in terms of CO2 mitigation, high levels of alkalis, sulfate, and chloride limit the extent to which CKD can be recycled in cement manufacturing (Al-Bakri et al., 2022). Carbonation of kiln dusts, involving the reaction of metal oxides with CO2 to form solid carbonates, has been proposed as an alternative CO2 sequestration method (Huntzinger et al., 2009; Adekunle, 2024):

$$CaO + CO_2 \rightarrow CaCO_3 \tag{7}$$

This process captures 1 mol CO2 per mol metal oxide, which is less than what can be achieved via CaO hydration and subsequent Ca(OH)2 dissociation in seawater (~1.68 mol CO2 mol-1 metal oxide). In landfills, both processes naturally occur when kiln dust is exposed to rainwater (Sreekrishnavilasam et al., 2006). However, limited water availability in large kiln dust piles promotes secondary precipitation of carbonates or clay minerals, reducing the effective CO2 sequestration. As such, the ad hoc CDR effect that occurs during landfill disposal of LKD and CKD remains uncertain. Similarly, application of kiln dust to agricultural soils for enhanced weathering purposes (as an alternative to primary mined rocks such as basalt or dunite) could contribute to CO2 removal, though restricted water availability may again increase the risk of secondary mineral formation (Buckingham and Henderson, 2024; Xu and Reinhard, 2025). The focus of this study is the usage of kiln dusts via OAE in natural marine environments.

Alternatively, kiln dusts could also be used in reactor-based OAE approaches, such as accelerated weathering of limestone (Rau and Caldeira, 1999; review in Huysmans et al., 2025). These methods allow fast, controlled, and easily monitored alkalinity addition, but require higher energy inputs and dedicated infrastructure compared to ship-based distribution in natural environments (Rau and Caldeira, 1999; Rau et al., 2007; Huysmans et al., 2025).

---

## Author Comment (AC3)

**Referee #3**

**General comments**

*The manuscript investigates the potential for atmospheric $CO_2$ removal via ocean alkalinity enhancement using waste products from cement and lime kilns. The study is based on laboratory dissolution experiments and evaluates the dissolution kinetics, $CO_2$ sequestration potential, and ecological risks associated with cement kiln dust (CKD) and lime kiln dust (LKD). In my opinion, the manuscript offers a valuable contribution to scientific progress within the scope of Biogeosciences, presenting new concepts and ideas for $CO_2$ removal that worth further testing and investigation.*

*Overall, the manuscript is well structured and clearly written. The language is fluent, the figures are clear and easy to interpret, and the amount and quality of the supplementary material are appropriate.*

*The conclusions are generally supported by the experimental results. However, I have reservations about the representativeness of the reported values, given that the experiments were conducted over relatively short timescales (8 hours and 15 days) in 200-mL polystyrene vials filled with filtered seawater under controlled laboratory conditions. These constraints limit the extent to which the findings can be extrapolated to real-world applications. In my view, the discussion, conclusions, and abstract should adopt a more cautious tone regarding the scalability and environmental impacts of the results.*

*The manuscript also addresses the potential ecological impacts of kiln dust dissolution. However, the assessment focuses primarily on turbidity and trace metal concentrations, without sufficiently considering biological responses or broader ecological consequences of kiln dust deployment in marine environments.*

*Finally, the Methods section could be strengthened by providing additional detail to facilitate reproducibility by other researchers.*

*I hope the comments below help the authors improve the clarity, rigor, and reproducibility of the study.*

We sincerely thank the reviewer for their thoughtful and supportive comments on our manuscript. The feedback has significantly improved the clarity, organization, and overall quality of the paper. In the sections below, we address each of their points and outline the corresponding revisions. Our responses are presented in regular black text, while the revised manuscript text appears in blue with changes highlighted in yellow.

We agree with the reviewer's general comments and have incorporated their suggestions throughout the manuscript, as detailed in the specific comments section below. In response to the recommendation that the discussion, conclusions, and abstract should adopt a more cautious tone, given that the study was conducted under controlled laboratory conditions, we have revised the abstract accordingly. Revisions to the discussion and conclusions are described under the relevant specific comments. The revisions made to the abstract are as follows:

Based on current industrial production rates, this translates into global CDR potentials of up to 8.7 ± 0.6 Mt $CO_2$ yr$^{-1}$ for LKD and 25 ± 2 Mt $CO_2$ yr$^{-1}$ for CKD. These estimates suggest that both materials could be viable OAE feedstocks, although further testing under conditions that more closely mimic natural coastal conditions is needed.

**Specific comments**

**Material and methods**

**- Please specify the source of all materials used in the experiments (e.g., supplier, kiln type, facility, geographical origin). This information is essential for assessing the broader applicability of the results and ensuring reproducibility.**

Due to a non-disclosure agreement, we are unable to provide the name of the kiln dust supplier or specific details regarding its production process. However, we respectfully disagree that this specific information is essential for evaluating broader applicability or ensuring reproducibility. The comprehensive physicochemical characterization included in our study provides all necessary information for comparing our results with future work using kiln dust from other sources, thereby enabling meaningful assessment and replication.

**- Please clarify which analytical methods were applied in each of the experiments. For Experiment II, indicate how total alkalinity (AT) was measured.**

The same analytical methods were used for both experiments, which are described in section 2.3 to avoid repetition. To improve clarity, we have now added a reference to this section within the methods description of Experiment II. The revised text now reads as follows:

Duplicate samples for dissolved inorganic carbon (DIC), dissolved metals, turbidity, and $A_T$ analysis were collected on both sampling days by drawing water with a syringe after opening the vials (analytical procedures are described in Sect. 2.3).

We have also added a reference to the section describing the solid-phase analyses to clarify that the same SEM–EDX procedure was applied to the recovered samples from Experiment II as to the fresh kiln dust.

The remaining suspension in the incubation vials was filtered through a 0.2 μm polycarbonate membrane filter to collect solids, which were rinsed with deionized water and then oven dried at 40°C in preparation for SEM-EDX analysis (see Sect. 2.1).

**- It appears that only two sampling times (after 1 day and after 15 days) were analyzed for Experiment II. Please clarify in the manuscript.**

For Experiment II, two sampling times were indeed selected. To improve clarity, this information, which was originally placed mid-way through the Experiment II description, has now been moved to the beginning. Furthermore, the term "longer-term" has been removed from the text in accordance with a later comment.

To assess the alkalinity generation potential and the possibility of secondary mineral formation, we conducted a second dissolution experiment with incubation periods of one and 15 days. The one-day (i.e. 24 h) incubation ensured complete dissolution of the reactive phases in the kiln dusts, while the 15-day incubation allowed for the verification of secondary mineral precipitation, in case this would occur.

**Lines 90-92 – In the sentence "Based on preliminary tests, three different masses of CKD and LKD were added targeting a specific aragonite saturation state (ΩArg) at the end ..." To facilitate reproducibility, please indicate the correspondence masses values (in grams) added to the 200 mL of FSW.**

This information was already provided in Table 1, but has now also been added to the text to clarify the kiln dust concentrations and their corresponding targeted aragonite saturation states:

Based on preliminary tests, three concentrations of CKD (30, 130, and 309 mg kg$^{-1}$) and LKD (11, 48, and 113 mg kg$^{-1}$) were selected to target different aragonite saturation states ($\Omega_{Arg}$ = 3.6, 5.7 and 9.7) (Table 1).

*Lines 103- 104- Referring to the 15-day experiment as a "long-term dissolution experiment" may be misleading in the context of OAE. I suggest referring simply to a 15-day experiment, unless additional justification is provided.*

We agree, the wording "longer-term" has been removed from the text (see revised text in previous comments).

*Lines 105-109 – For reproducibility, please provide details on the number of replicates used and the number of vials per treatment.*

Thank you for noting that this crucial info was missing. The number of replicates has now been added to the text.

Vials were closed tightly and had minimal headspace to minimize gas exchange with the atmosphere. Experiment II was conducted in triplicate at ambient room temperature (17.5–22.7 °C).

*Line 104 – Are 200 mL plastic vials representatives of sea water collum? Please explain the rationale for choosing this container size.*

We agree with the reviewer's later comments that larger-scale experiments, which better mimic natural conditions, are ultimately needed to fully assess the CDR potential and ecological risks of kiln-dust-based OAE. However, the use of 200 mL vials in this study was intentional: small-scale incubations allow rapid, cost-effective screening of multiple treatments simultaneously, which is appropriate for a first assessment of whether a material is suitable for OAE. This has been clarified in the text as follows:

Kiln dusts were weighed in small aluminium (Al) foil cups using a micro balance (XP26 Excellence Plus, Mettler Toledo) and then transferred to 200 mL polystyrene vials with polyethylene screw caps containing approximately 200 mL of FSW. These small-scale laboratory experiments provide a high-throughput, cost-effective first assessment of a material's suitability for OAE.

*Line 109 – Please clarify why different rotation speeds were used in the two experiments (700 rpm in Experiment I vs. 14 rpm in Experiment II). What was the intended effect of this difference?*

Different rotation speeds were used due to differences in the experimental set-ups. In Experiment I, vials were kept upright on a magnetic stirrer to allow continuous pH monitoring. In Experiment II, bottle rollers were employed since pH monitoring was not required and this setup allowed simultaneous incubation of a larger number of vials. In both experiments, rotation speeds were selected to ensure effective mixing and to maintain particles in suspension, creating ideal conditions for dissolution and enabling assessment of the maximum CDR potential. The rationale for these rotation speeds is further clarified in the text, which now reads as follows:

Seawater temperature was kept constant at 20°C during the incubation by means of a water bath (T100, Grant). Magnetic stirring was applied at a rate of 700 rotation per minute (RPM) to ensure good mixing of the suspension and to create optimal dissolution conditions.

Experiment II was conducted in triplicate at ambient room temperature (17.5–22.7 °C). Vials were subsequently incubated on bottle rollers (ThermoFisher Scientific) for 1 or 15 days at 14 RPM, a speed sufficient to keep particles suspended and ensure optimal dissolution conditions.

***Lines 109-110 – In the sentence "The one-day incubation reflects the time needed for complete dissolution of the reactive phases in the kiln dusts.", please specify the exact duration in hours from the start of the experiment until dissolution was considered complete.***

The exact duration of complete reactive phase dissolution is not known, but Experiment I suggests it occurs within 24 hours, a finding confirmed by the 15-day incubation, which showed no additional alkalinity release compared to the 1-day incubation of Experiment II. The sentence referred to by the reviewer has been rephrased to:

The one-day (i.e. 24 h) incubation ensured complete dissolution of the reactive phases in the kiln dusts, while the 15-day incubation allowed for the verification of secondary mineral precipitation, in case this would occur.

***Lines 237-238 – In the sentence "…while for LKD, the ΔAT curve showed a maximum at higher concentrations (Fig. 2B)" does not seem to match the figure: the maximum appears to occur before the highest concentration. Please revise accordingly.***

You are correct; the sentence has been revised as follows:

After one day, $\Delta A_T$ showed a monotonous increase with the CKD concentration, while for LKD, the $\Delta A_T$ curve reached a maximum at 69 mg kg$^{-1}$ and decreased at higher application concentrations (Fig. 2B).

***Lines 318-319 – Regarding this sentence "The compositional complexity of kiln dusts underscores the need for detailed mineralogical and chemical characterization to properly assess the CDR potential and environmental risks in OAE applications.", more details about the provenance of the kiln dust should be added to the methods section.***

As mentioned previously, we are unable to disclose the origin of the kiln dusts; however, we believe this information is not critical given the detailed physicochemical characterization provided.

***Line 363 – In the sentence "… leading to a net AT loss, as seen at the highest LKD concentration after 15 days (Fig. 2A)". the reference appears to be to Figure 2B rather than Figure 2A, please check.***

You are correct, we changed the reference from Figure 2A to 2B.

***Line 371 – In the sentence "… several minutes during ship-based ocean liming Caserini et al. (2021), which may have an impact on marine life if pH exceeds". It seems that the reference should be placed between brackets, please check.***

The reference should indeed be between brackets, this has now been corrected.

***Line 373-374 – "To avoid temporary exceedances of pH 9, CKD concentrations should stay below 343–502 mg kg-1, and LKD below 102–149 mg kg-1, depending on local seawater conditions (AT = 2350 µmol kg-1, DIC = 2100 µmol kg-1, salinity = 35, temperature = 10–25 °C)."***

***Your experiment was conducted at a salinity level of 32.3 ± 0.5 and a temperature level of 17.5–22.7 °C. Are these values comparable to a salinity of 35 and a temperature of 10–25 °C? Would these variations in salinity and temperature influence the concentrations of CKD and LKD? What impact would they have on the pH? These points need to be clarified in the manuscript.***

Temperature, salinity, hydrostatic pressure, and the background seawater total alkalinity and DIC all influence seawater pH, and thus the amount of kiln dust that can be added before reaching pH 9. The values reported in the manuscript correspond to average surface seawater conditions under temperate and tropical temperatures, calculated from the maximum specific alkalinity release measured for LKD (8.0 mmol g$^{-1}$) and CKD (2.4 mmol g$^{-1}$). Using these specific alkalinity release rates, local maximum application concentrations can be readily calculated once the seawater conditions at a specific deployment site are known. This clarification has now been added to the text as follows:

To avoid temporary exceedances of pH 9, CKD concentrations should stay below 343–502 mg kg$^{-1}$, and LKD below 102–149 mg kg$^{-1}$ under average surface seawater conditions ($A_T$ = 2350 µmol kg$^{-1}$, DIC = 2100 µmol kg$^{-1}$, salinity = 35, temperature = 10–25 °C). Application concentrations must be further tailored to local seawater geochemistry at the deployment site to prevent exceeding the pH 9 threshold.

***Line 385-387 – "In real applications, kiln dust will be rapidly mixed into much larger volumes of surface water, …."***

***How representative is your experiment, that used 200 ml sea-water bottle, to be extrapolated for real application conditions? How would be guarantee a concentration below the referenced concentrations at the discharge point? In my opinion, without stronger justification, the scalability of the results remains uncertain.***

Our results demonstrate a clear linear relationship between suspended particle concentrations and seawater turbidity for the kiln dust studied. To ensure concentrations remain below turbidity guidelines, numerical modelling is required that incorporates local hydrodynamic conditions and kiln dust particle behaviour to determine appropriate discharge rates from the ship. We have elaborated on this in the text and cited two studies addressing modelling considerations for OAE and the effective settling of mineral particles in the ocean relevant to mCDR applications.

In real applications, kiln dust will be rapidly mixed into much larger volumes of surface water, meaning that the allowable concentration in the input stream will depend on the discharge rate, intensity of local turbulence, and kiln dust settling time (which is primarily determined by initial particle size). Accurate numerical modelling to determine suitable discharge rates therefore requires detailed knowledge of the environmental conditions at the deployment site and the behaviour of kiln dust particles under varying hydrodynamic conditions (Fennel et al., 2023; Yang and Timmermans, 2024). This information is essential to extrapolate small-scale laboratory results to realistic field scenarios and ensure that concentrations remain below guideline levels.

***Line 407-409 – "Assuming full mixing in the top 10 cm of the sediment, up to 1.4 kg CKD or 74.8 kg LKD per m² could be applied ….."***

***Please clarify what would be the resulting thickness of kiln dust deposited on the seafloor? Even if the 10-cm surface layer is assumed to be fully mixed over time, the initial deposition could create a substantial layer of fine material. Benthic infauna and epifauna cannot survive rapid burial under more than ~1–2 cm of sediment, and a thick layer of fine particles would also strongly reduce oxygen exchange at the sediment–water interface. This physical disturbance should also be addressed in the discussion, as compliance with chemical SQGs alone does not ensure ecological safety.***

We agree that physical disturbance is an important aspect that should be incorporated into the discussion. The resulting thickness of a deposited kiln-dust layer depends on its bulk density and solid-phase density, which vary with grain morphology, mineralogy, and local grain packing. Assuming a kiln

dust bulk density of 0.62 g cm$^{-3}$ based on the study by Nikolov et al. (2025), the maximum application rates derived from trace-metal SQGs would produce surface layers of approximately 0.2 cm for 1.4 kg m$^{-2}$ CKD and 12 cm for 74.8 kg m$^{-2}$ LKD, assuming uniform spreading.

These estimates indicate that physical burial is of low risk during CKD spreading, whereas for LKD the potential for burial impacts is considerable. A layer of this thickness would be expected to cause mortality of low-mobility benthic fauna and strongly reduce oxygen exchange at the sediment–water interface, thereby altering sediment geochemical processes. However, from a CDR perspective, applying such a thick layer of fine-grained LKD would also be inadvisable for other reasons, including the reduced efficiency of CaCO$_3$ dissolution at high application rates (Dale et al., 2024) and potential changes in habitat suitability for benthic organisms (Flipkens et al., 2024).

We have now incorporated the estimated kiln-dust layer thicknesses and associated physical impacts into the revised manuscript text as suggested.

Assuming full mixing in the top 10 cm of the sediment, up to 1.4 kg CKD or 74.8 kg LKD per m$^2$ could be applied to pristine sediments without exceeding the strictest marine SQG of 30.2 mg kg$^{-1}$ for Pb (Appendix C). Using a kiln dust bulk density of 0.62 g cm$^{-3}$ (Nikolov et al., 2025), this corresponds to an applied layer of approximately 0.2 cm for CKD and 12 cm for LKD. The exact layer thickness would of course be dependent on the specific kiln dust properties and local grain packing. These estimates indicate that burial risk for benthic organisms is small for CKD, but could be considerable for LKD, since the deposition of a cm-thick layer could substantially impact the resident benthic infauna and epifauna. Moreover, applying LKD at this scale is also not advisable because it may lead to changes in habitat suitability (e.g., grain size, permeability, organic carbon content) (Speybroeck et al., 2006; Flipkens et al., 2024) and alter geochemical sediment processes (see Sect. 4.4). Overall, these findings underscore the need for ecotoxicological testing and cautious application of kiln dust to avoid ecological harm.

*Line 420-424 – The manuscript notes that large-scale fining of permeable sediments by kiln dust could reduce oxygen penetration depth and thereby limit the zone of metabolic CaCO$_3$ dissolution. However, the biological implications of such physical and geochemical changes are not discussed. What would be the expected impact on benthic organisms living within these sediments, particularly those that produce or maintain calcium carbonate shells (e.g., molluscs, foraminifera, small crustaceans)? Reduced permeability, shallower oxic layers, and burial by fine material could harm calcifiers through smothering, reduced oxygen availability, and altered porewater chemistry. I recommend addressing these ecological consequences alongside the geochemical considerations*

We have expanded the discussion to address additional geochemical and physical changes beyond reduced oxygen penetration depth. We also added a reference to Section 4.3, where potential ecological impacts, including burial and changes in habitat suitability arising from both physical and chemical alterations, are now discussed as mentioned in the previous comment. The revised text reflects these broader considerations.

However, large-scale fining of sediment with kiln dust could reduce sediment properties, such as the permeability, solute exchange rates, and oxygen penetration depth (Speybroeck et al., 2006; Ahmerkamp et al., 2017). The latter would limit the zone in which metabolic CaCO$_3$ dissolution can occur. Relatively high CaCO$_3$ concentrations may further reduce the dissolution efficiency (i.e. dissolution rate per amount of CaCO$_3$ added) (Dale et al., 2024). Additional ecological impacts may arise through changes in the physical texture of the solid sediment matrix and modifications of porewater conditions (see Sect. 4.3). The potential for enhanced sedimentary alkalinity generation via residual kiln dust addition to organic-rich, carbonate-poor marine sediments therefore warrants further experimental investigation.

***Line 470-471 – "Overall, LKD and, to a lesser extent, CKD show promise for OAE, with a CDR potential of up to 13.4 Mt year-1 for LKD and 57 Mt year-1 for CKD at current production levels"***

***In my opinion, given the significant uncertainties surrounding ecological impacts, including water-column responses, sediment interactions, and potential biological effects, the Conclusions should acknowledge these uncertainties and reflect them in the overall assessment of kiln dust as a viable CDR strategy.***

We agree that the conclusions could be more carefully worded, given that this is the first study assessing the potential of kiln dust for OAE and that our experiments were conducted on a small laboratory scale. Accordingly, we have revised the text to reflect these uncertainties and included recommendations for future research to address water column responses, sediment interactions, and potential biological impacts under more realistic environmental conditions.

Overall, LKD and, to a lesser extent, CKD show promise for OAE, with a CDR potential of up to 13.4 Mt year$^{-1}$ for LKD and 57 Mt year$^{-1}$ for CKD at current production levels, based on our small-scale laboratory experiments. However, additional experiments that more closely mimic natural conditions are warranted to further constrain particle behaviour in the water column, interactions with sediments, and potential biological impacts of kiln-dust–based OAE.

**References:**

Ahmerkamp, S., Winter, C., Krämer, K., Beer, D. d., Janssen, F., Friedrich, J., Kuypers, M. M., Holtappels, M.: Regulation of benthic oxygen fluxes in permeable sediments of the coastal ocean, Limnol. Oceanogr., 62, 1935-1954, https://doi.org/10.1002/lno.10544, 2017.

Dale, A. W., Geilert, S., Diercks, I., Fuhr, M., Perner, M., Scholz, F., Wallmann, K.: Seafloor alkalinity enhancement as a carbon dioxide removal strategy in the Baltic Sea, Commun. Earth Environ., 5, 452, https://doi.org/10.1038/s43247-024-01569-3, 2024.

Fennel, K., Long, M. C., Algar, C., Carter, B., Keller, D., Laurent, A., Mattern, J. P., Musgrave, R., Oschlies, A., Ostiguy, J.: Modeling considerations for research on Ocean Alkalinity Enhancement (OAE), State of the Planet Discussions, 2023, 1-47, https://doi.org/10.5194/sp-2-oae2023-9-2023, 2023.

Flipkens, G., Dujardin, V., Salden, J., T'Jollyn, K., Town, R. M., Blust, R.: Olivine avoidance behaviour by marine gastropods (Littorina littorea L.) and amphipods (Gammarus locusta L.) within the context of ocean alkalinity enhancement, Ecotoxicol. Environ. Saf., 270, 115840, https://doi.org/10.1016/j.ecoenv.2023.115840, 2024.

Nikolov, A., Kostov-Kytin, V., Tarassov, M., Tsvetanova, L., Jordanov, N. B., Karamanova, E., Rostovsky, I.: Characterization of cement kiln dust from Bulgarian cement plants, J. Chem. Technol. Metall., 60, 455-463, https://doi.org/10.59957/jctm.v60.i3.2025.11, 2025.

Speybroeck, J., Bonte, D., Courtens, W., Gheskiere, T., Grootaert, P., Maelfait, J. P., Mathys, M., Provoost, S., Sabbe, K., Stienen, E. W.: Beach nourishment: an ecologically sound coastal defence alternative? A review, Aquat. Conserv.: Mar. Freshw. Ecosyst., 16, 419-435, https://doi.org/10.1002/aqc.733, 2006.

Yang, A. J., Timmermans, M.-L.: Assessing the effective settling of mineral particles in the ocean with application to ocean-based carbon-dioxide removal, Environ. Res. Lett., 19, 024035, https://doi.org/10.1088/1748-9326/ad2236, 2024.

---

## Author Comment (AC4)

**Referee #2**

*Overview:*

*The manuscript from Flipkens et al. explores the suitability of two by-products of the cement industry, cement and lime kiln dusts (CKD and LKD, respectively). Using laboratory experiments, they report first estimates on the dissolution kinetics of such feedstocks in natural seawater, both using short- and long-term experiments, while expanding on the risks for $CaCO_3$ precipitation and potential trace metals release. Finally, CDR estimates are given, assuming the use of all available CKD and LKD, and considering inorganic thresholds and water quality and safety guidelines.*

*Overall, I found the manuscript very enjoyable to read. While some people may express reluctance to consider such by products for OAE, their significant alkalinity release makes them suitable candidates, and the presented data support this. The introduction is rather comprehensive, and most required information is already provided. The material and method section is also mostly complete with some smaller comments pointed out below. The results section is rather dense, yet all required information is well presented. Finally, the discussion is convincing and reports well on the implication of kiln dust based OAE. Some further points could be added to the discussion, especially when it comes to the alkalinity generation. While LKD was following estimates, the nearly doubling alkalinity generation from CKD is yet to be fully addressed. I believe a more detailed discussion could benefit the paper. It would be beneficial to try and characterise the amorphous phases that clearly seem to be the responsible factor. I do not request further analyses but rather consider whether using the elemental composition of CKD (Table A1), an estimate of the amorphous phases' composition could be derived. Another point that could benefit from further interpretation is the fate of KD and especially the potential secondary minerals precipitated. It is clear that non soluble phases such as $CaCO_3$ would eventually sink and as described, potentially dissolve on the seabed. It would also be the case for secondary $CaCO_3$ formed from the high alkalinity generated. One could consider discussing the dissolution on the seabed of both the $CaCO_3$ from KD and the freshly precipitated $CaCO_3$ in the water column. The overall potential would probably increase, and if it is significant, it would be worth mentioning. Finally, a smaller point regarding the turbidity discussion. Discussing such environmental guideline is a great idea and of high importance. Unless I am mistaken, the turbidity is reported for 24h. In line 343, it is mentioned that the KD particles would remain in the surface ocean mixed layer for about one hour. Therefore, one could safely assume that after 1h the turbidity is back to low enough levels that new KD could be added. Under such assumption, the overall KD release could be increased, ultimately increasing the CDR potential. If it is correct, it could be quicky discussed.*

*Considering the minor comments mentioned above, and once they have been addressed, I am fully supporting the publication of the manuscript.*

We thank the reviewer for their insightful feedback and thorough review of the manuscript. In response to your first suggestion, we have expanded the discussion regarding the alkalinity generation from CKD, which was higher than expected based on the mineralogical composition. By comparing the major elemental composition determined via ICP-OES with that derived from crystalline phases identified by XRD, we now attribute the excess to the potential presence of amorphous calcium aluminosilicates, alkali sulfates, and poorly crystalline lime phases. These phases could possibly explain a significant portion of the previously unexplained alkalinity release. Additionally, as suggested by Reviewer 1, we now discuss the potential contribution of calcium silicates to alkalinity generation in CKD (changes in the manuscript text (blue) are highlighted in yellow below).

In LKD, alkalinity release was fully attributed to the dissolution of portlandite ($Ca(OH)_2$) and lime ($CaO$), whereas in CKD these phases explained only about half ($54 \pm 3$ %) of the observed alkalinity release. Calcium silicates (e.g. larnite $Ca_2SiO_4$) are also alkalinity-generating phases that occur in minor amounts in CKD. They originate form the raw materials used in cement production (e.g. iron ore, clay, or shale) and exhibit a relatively high reactivity in water (Brand et al., 2019; Adekunle, 2024). Dissolution of the larnite present in our CKD sample (~ 2.1 %) could account for $17 \pm 1$% of the observed $A_T$ release, which hence provides a substantial additional contribution. The remaining ~29% of the alkalinity released from CKD likely originated from dissolution of amorphous phases, including (partially dehydrated) clay minerals, reactive amorphous silica, and kiln-derived materials such as fly ash or slag (Khanna, 2010; Pavía and Regan, 2010). ICP-OES analysis revealed 8.0 % Ca, 2.7 % K, 0.7 % Na, 0.9 % S, 1.6% Fe and 3 % Al that were not accounted for by the crystalline phases detected via XRD (Table 2 and Table A1). This suggests the possible presence of amorphous calcium aluminosilicates, alkali sulfates, and poorly crystalline $CaO$ or $Ca(OH)_2$, which may have contributed to the remaining alkalinity upon dissolution in seawater (Hu et al., 2024; Nikolov et al., 2025).

Furthermore, we adjusted the wording in line 197 from "congruent" to "generally in line" to soften the phrasing and clarify that the element compositions measured via ICP-OES and those derived from XRD were not an exact match.

The observed elemental composition was generally in line with the XRD results, showing high calcium contents in both CKD (27.8 wt%) and LKD (44.9 wt%), which fall within the range previously reported for CKD (14–46 %) and LKD (20–49 wt%) (Collins and Emery, 1983; Pavía and Regan, 2010; Latif et al., 2015; Drapanauskaite et al., 2021; Dvorkin and Zhitkovsky, 2023).

We thank the reviewer for the valuable suggestion to discuss the potential dissolution of secondary $CaCO_3$ precipitates after they settle and become buried in the sediment. We have now addressed the possible contribution of both $CaCO_3$ included in kiln dust as well as freshly precipitated secondary $CaCO_3$ for alkalinity generation through subsequent dissolution on the seabed. This has been incorporated into the manuscript as follows:

Prolonged exceedance of critical saturation thresholds can trigger "runaway $CaCO_3$ precipitation", leading to a net $A_T$ loss, as seen at the highest LKD concentration after 15 days (Fig. 2A) (Moras et al., 2022). Under natural conditions, freshly precipitated aragonite may redissolve after dilution in the ship's wake, especially when not yet fully crystallized, recovering some of the lost alkalinity due to secondary precipitation (Hartmann et al., 2023). Aragonite precipitates that settle onto the sediment at the deployment site may undergo further metabolic dissolution provided that geochemical conditions are favourable, offsetting the earlier alkalinity loss (see Section 4.4). However, these fine-grained precipitates could also disperse far from the deployment site, thus complicating monitoring, reporting, and verification (MRV) of CDR via kiln-dust-based OAE. So, despite that some secondary aragonite may redissolve, its formation is best minimized to maximize the alkalinization potential. Based on our temporal dissolution data (Fig. 1A-B), it is recommended to adjust the OAE dispensing and deployment procedure in such a way, that dilution to $\Omega_{Arg} < 5$ occurs within minutes as to minimize secondary mineral precipitation.

Furthermore, we added a statement in the section on the longer term fate of unreacted phases to note that secondary $CaCO_3$ precipitates could undergo the same fate as the residual calcite in the kiln dust.

In coastal and shelf environments, this residual material would rapidly settle to the seafloor. The residual fraction consists primarily of $CaCO_3$ phases (52 % in CKD and 72 % in LKD). When residual $CaCO_3$ or freshly precipitated secondary $CaCO_3$ become mixed into the seabed through local

hydrodynamics and bioturbation, porewater acidification resulting from microbial degradation of organic matter can trigger metabolic $CaCO_3$ dissolution (Rao et al., 2012; Kessler et al., 2020).

The turbidity guidelines indeed specify values that should not be exceeded for more than 24 hours. The wording in line 343 was incorrect: 85% of the LKD particles and all CKD particles would remain suspended in a 200 m water column for at least one hour, not "about" one hour. This has now been changed in the text.

Using our measured particle size distribution and assuming particle sinking follows Stokes' law, all CKD particles and the majority of LKD particles (85 ± 2 % V/V) will remain in the surface ocean mixed layer (assumed to be 200 m) for at least one hour, thus allowing sufficient time for most reactive phases to dissolve and generate alkalinity (Appendix A Sect. A2).

We do agree that particle settling time is a key factor in determining how frequently kiln dust can be applied without exceeding turbidity guidelines. This consideration has now been added to the sentence noting that application rates should be adjusted according to ship discharge rates and local hydrodynamics.

In real applications, kiln dust will be rapidly mixed into much larger volumes of surface water, meaning that the allowable concentration in the input stream will depend on the discharge rate, the intensity of local turbulence, and the kiln dust settling time (which is primarily determined by particle size).

***Comments:***

***Line 6: I am not sure whether "ocean liming" is a suitable keyword here, considering it investigates kiln dust potential for OAE rather than lime. But this can be up to the authors.***

We understand the potential confusion. We included "ocean liming" as a keyword because the primary alkalinity-generating phases in the kiln dust are quicklime (CaO) and hydrated lime ($Ca(OH)_2$). That said, other phases are present as well, so it is indeed debatable whether kiln dust perfectly aligns with the conventionally used definition of ocean liming, such as the one by Renforth et al. (2013), who describe Ocean Liming as "the addition of alkalinity in the form of calcium or magnesium oxide/hydroxide (CaO or MgO) to increase ocean pH and $CO_2$ uptake." Nevertheless, we believe this research is highly relevant to the ocean liming community. Lime kiln dust is a by-product of lime production, and its potential use for OAE should be recognized, especially since it could possibly increase the overall CDR potential per tonne of calcined limestone when considered alongside the produced lime. For these reasons, we prefer to retain "ocean liming" as a keyword for the paper.

***Line 34: I believe brackets are missing and should read "brucite $(Mg(OH)_2)$".***

Indeed, brackets have been added.

***Line 42: one could argue the use of "rapid" here, as it can take month to years before the alkalised water equilibrates with atmospheric $CO_2$. Please edit accordingly.***

We agree that the word rapid was misplaced. Although the release of alkalinity is rapid, the subsequent atmospheric $CO_2$ drawdown indeed occurs more slowly. Therefore, we replaced $CO_2$ sequestration with seawater alkalinization, so the sentence now reads as follows:

Ocean liming has the benefit of rapid seawater alkalinization upon deployment, and offers the potential to remove gigatons of atmospheric $CO_2$ annually, with ample global reserves to support deployment (Caserini et al., 2022; Foteinis et al., 2022).

***Line 57: consider moving "hence" at the start of the sentence.***

"hence" was moved to the start of the sentence.

***Line 69: has this method been tested and proven to be reliable and efficient at dissolving such product? If so, a reference (or in-house quality control) could be inserted.***

The HF–HClO$_4$–HNO$_3$ digestion is a strong total digestion method and is widely used to dissolve refractory and silicate-rich matrices. The fact that measured major element concentrations slightly exceed those inferred from XRD for the crystalline phases indicates that the digestion effectively dissolved the crystalline material, as well as the amorphous and poorly crystalline fractions, which are expected to dissolve even more readily.

In-house quality control (blanks, duplicates, and certified reference materials (CRM)) was performed. Although the CRM used (river clay ISE 921) differs in matrix from our sample, recoveries were within 98–109%, supporting the efficiency and consistency of the digestion. Additional information on the CRM and elemental recovery ranges have now been added to the text.

Quality control measures included a blank, two certified river clay standards (ISE 921), and a duplicate sample. Recorded elemental concentration were between 98 % and 109 % of certified values.

***Lines 87-88: how long was the seawater aerated for? Was there any check to confirm equilibration? (pH measurements, pCO$_2$, etc.).***

The seawater was aerated for 24 hours, as described in the text. For Experiment I, both pH$_T$ and alkalinity of the initial seawater were measured, while for Experiment II, alkalinity and DIC were analyzed, allowing calculation of pCO$_2$. Values ranged from 458 to 557 µatm, indicating that seawater pCO$_2$ remained a little above the atmospheric level (~420 µatm). This has now been clarified in the text as follows:

Filtered (<0.2 µm) seawater from the Eastern Scheldt (saline water body in The Netherlands adjacent to the North Sea; salinity 32.3 ± 0.5) was obtained from Stichting Zeeschelp (Kamperland, The Netherlands). The filtered seawater (FSW) was aerated for 24 hours before use, yielding an initial seawater pCO$_2$ of 458–557 µatm.

***Line 111: how were DIC samples taken? Given the gas sensitivity of these samples, it would be interesting to report whether a peristaltic pump was used or any other apparel.***

DIC samples were collected by simply opening the vials and withdrawing water with a syringe. To minimize atmospheric exposure, DIC samples were always taken first, before any other analytes. This has now been clarified in the text as follows:

Duplicate samples for dissolved inorganic carbon (DIC), dissolved metals, turbidity, and A$_T$ analysis were collected on both sampling days by drawing water with a syringe right after opening the vials (analytical procedures are described in Sect. 2.3). DIC samples were collected first to minimize the exposure time to the atmosphere.

Given that samples were taken over a timescale of only a few minutes from stagnant water, the uptake of atmospheric CO$_2$ during sampling is expected to be very small. A more important source of CO$_2$ exchange likely stemmed from the small headspace that unavoidably forms when closing any vial after normal filling. This headspace allows CO$_2$ to equilibrate with the sample over the full duration of the incubations (1 or 15 days), causing DIC increases that partially mask DIC losses through secondary mineral precipitation at higher kiln dust application levels.

This CO₂ uptake influenced the results in both experiments. In Experiment I, it caused a decrease in seawater pH and an underestimation of the fraction of dissolved reactive phases, as discussed in Lines 350–354. In Experiment II, the focus was on alkalinity generation and potential secondary mineral precipitation. While $A_T$ is not directly affected by atmospheric $CO_2$ exchange, DIC is affected. As a result, the reported aragonite saturation state is likely lower than the true peak values during the incubations, which may influence the inferred critical aragonite saturation threshold. Moreover, we cannot quantitatively estimate the amount of $CaCO_3$ that precipitated accurately because the observed DIC loss reflects both secondary mineral precipitation and $CO_2$ uptake from the small vial headspace. Despite these uncertainties, the derived critical aragonite saturation state ($\Omega_{Arg} \approx 5$) aligns well with your previous findings using lime, suggesting that any resulting error is likely small.

Moreover, under natural conditions, atmospheric $CO_2$ drawdown would occur alongside alkalinization, though more slowly, progressively altering aragonite saturation levels. To highlight this ongoing process, and to clarify that our experimental setup was not designed to study completely unequilibrated solutions, we added the following text to the manuscript:

Duplicate samples for dissolved inorganic carbon (DIC), dissolved metals, turbidity, and $A_T$ analysis were collected on both sampling days by drawing water with a syringe after opening the vials (analytical procedures are described in Sect. 2.3). DIC samples were collected first to minimize exposure time to the atmosphere. Nevertheless, some $CO_2$ exchange inevitably occurred during the incubations because the vials contained a small headspace, meaning the solutions could partially re-equilibrate with the air. However, this also reflects natural deployment conditions, where atmospheric $CO_2$ exchange occurs alongside alkalinization, although at a slower rate.

**Line 125-126: since the pH has been measured and later reported on the total scale, it would be beneficial to report it as $pH_T$ throughout the text**

We agree and have changed pH to $pH_T$ throughout the text.

**Line 150: salinity has no unit, please remove the "ppt". alternatively, you could report the ionic strength instead, but it is not required**

ppt has been removed from the text.

**Line 164: I believe PHREEQC provides saturation indexes, not saturation states. The term $\Omega$ is usually used only for $CaCO_3$ if I am correct**

You are correct that PHREEQC outputs saturation indices (SI), not saturation states ($\Omega$). To obtain saturation states, we converted the PHREEQC-derived SI values using $\Omega = 10^{SI}$. We have clarified this in the manuscript as follows:

Saturation index (SI) values for kiln dust mineral phases were calculated using PHREEQC Interactive (version 3.7.3-15968) with the LLNL thermodynamic database (Parkhurst and Appelo, 2013). Saturation indices were converted to saturation states ($\Omega$) according to $\Omega = 10^{SI}$.

Regarding the terminology, although $\Omega$ is most commonly associated with the saturation state of $CaCO_3$, the symbol can be used more broadly for other minerals as well. In this study, $\Omega$ is used to denote the saturation state of each mineral evaluated, not exclusively $CaCO_3$.

**Lines 164-169: I am not sure I understand correctly. First it is said that saturation states were calculated with PHREEQC (line 164). Then, it is said that $\Omega A$ and $\Omega C$ were not computed in PHREEQC. Why that?**

Thank you for pointing this out. To clarify, aragonite and calcite saturation states were not computed in PHREEQC, but calculated using the AquaEnv R package. This approach was chosen because most studies reporting CaCO₃ saturation states use Seacarb or CO2SYS, rather than PHREEQC for computation. The LLNL PHREEQC database employed in our study uses thermodynamic data from Plummer and Busenberg (1982) to calculate CaCO₃ saturation states, whereas Seacarb, AquaEnv, and CO2SYS generally use solubility product constants from Mucci (1983) and carbonic acid dissociation constants from Lueker et al. (2000).

Saturation states for other mineral phases present in the kiln dust could not be calculated in AquaEnv and were therefore derived using PHREEQC. The rationale for using AquaEnv to calculate CaCO₃ saturation, ensuring consistency with commonly used thermodynamic data and facilitating comparison with other studies, has now been added to the text.

Aragonite and calcite saturation states were not computed in PHREEQC but were instead calculated using the AquaEnv package in R, as described previously (Hofmann et al., 2010). AquaEnv uses carbonic acid dissociation constants from Lueker et al. (2000) and solubility product constants for CaCO₃ from Mucci (1983), which differ from the thermodynamic data used in the LLNL PHREEQC database to describe the carbonate system. We used the AquaEnv approach to remain consistent with the methodology commonly applied in most OAE studies.

**Line 246: I believe that "for CKD is missing after the 1.3 mmol g -1 .**

This was indeed missing and "for CKD" has now been added to the text. Furthermore, the number 1.3 has been changed by 1.7 to take into account potential additional alkalinity generation through calcium silicate dissolution as requested by reviewer 1.

The maximum specific alkalinity release for LKD ($8.02 \pm 0.53$ mmol $g^{-1}$) was more than three times higher than that of CKD ($2.38 \pm 0.16$ mmol $g^{-1}$). Moreover, the LKD value was in good agreement with the theoretical prediction (8.8 mmol $g^{-1}$; see section 3.1), while the CKD value deviated more substantially from the theoretical estimate (1.7 mmol $g^{-1}$).

**Line 264: it would be great to have the 0 on the y axis of tile B.**

We agree and have added 0 on the y-axis of the figure 2B.

**Lines 293-294: how can turbidity be slightly greater yet significantly greater? Do you mean statistically significantly greater?**

Yes, this indeed needs to be statistically significantly greater. The term "statistically" has now been added to the text.

**Line 310: it would be great to have the 0 on the y axis of tile B.**

We agree and have added 0 on the y-axis of the figure 4B.

**Line 330: I believe "be" should be deleted.**

Indeed, we removed "be" from the text.

**References:**

(1)     Brand, A. S., Gorham, J. M., Bullard, J. W., 2019. Dissolution rate spectra of β-dicalcium silicate in water of varying activity. *Cem. Concr. Res. 118*, 69-83. doi:https://doi.org/10.1016/j.cemconres.2019.02.014

(2)     Adekunle, S. K., 2024. Carbon sequestration potential of cement kiln dust: mechanisms, methodologies, and applications. *J. Clean. Prod. 446*, 141283. doi:https://doi.org/10.1016/j.jclepro.2024.141283

(3)     Hu, M., Dong, T., Cui, Z., Li, Z., 2024. Mechanical behavior and microstructure evaluation of quicklime-activated cement kiln dust-slag binder pastes. *Materials. 17* (6), 1253. doi:https://doi.org/10.3390/ma17061253

(4)     Nikolov, A., Kostov-Kytin, V., Tarassov, M., Tsvetanova, L., Jordanov, N. B., Karamanova, E., Rostovsky, I., 2025. Characterization of cement kiln dust from Bulgarian cement plants. *J. Chem. Technol. Metall. 60*, 455-463. doi:https://doi.org/10.59957/jctm.v60.i3.2025.11

(5)     Hartmann, J., Suitner, N., Lim, C., Schneider, J., Marín-Samper, L., Arístegui, J., Renforth, P., Taucher, J., Riebesell, U., 2023. Stability of alkalinity in ocean alkalinity enhancement (OAE) approaches–consequences for durability of CO 2 storage. *Biogeosci. Disc. 20* (4), 781-802. doi:https://doi.org/10.5194/bg-20-781-2023

(6)     Renforth, P., Jenkins, B., Kruger, T., 2013. Engineering challenges of ocean liming. *Energy. 60*, 442-452. doi:https://doi.org/10.1016/j.energy.2013.08.006

(7)     Plummer, L. N., Busenberg, E., 1982. The solubilities of calcite, aragonite and vaterite in CO2-H2O solutions between 0 and 90 C, and an evaluation of the aqueous model for the system CaCO3-CO2-H2O. *Geochim. Cosmochim. Acta. 46* (6), 1011-1040. doi:https://doi.org/10.1016/0016-7037(82)90056-4

(8)     Lueker, T. J., Dickson, A. G., Keeling, C. D., 2000. Ocean pCO2 calculated from dissolved inorganic carbon, alkalinity, and equations for K1 and K2: validation based on laboratory measurements of CO2 in gas and seawater at equilibrium. *Mar. Chem. 70* (1-3), 105-119. doi:https://doi.org/10.1016/S0304-4203(00)00022-0

(9)     Mucci, A., 1983. The solubility of calcite and aragonite in seawater at various salinities, temperatures, and one atmosphere total pressure. *Am. J. Sci. 283* (7), 780-799. doi:https://doi.org/10.2475/ajs.283.7.780